# Pharmacogenetics and Pain Treatment with a Focus on Non-Steroidal Anti-Inflammatory Drugs (NSAIDs) and Antidepressants: A Systematic Review

**DOI:** 10.3390/pharmaceutics14061190

**Published:** 2022-06-01

**Authors:** Farzin Zobdeh, Ivan I. Eremenko, Mikail A. Akan, Vadim V. Tarasov, Vladimir N. Chubarev, Helgi B. Schiöth, Jessica Mwinyi

**Affiliations:** 1Department of Surgical Sciences, Division of Functional Pharmacology and Neuroscience, Biomedical Center (BMC), Uppsala University, Husargatan 3, P.O. Box 593, 75124 Uppsala, Sweden; zobdeh_farzin@yahoo.com (F.Z.); eremenko_i_i@student.sechenov.ru (I.I.E.); akan.mikhail@mail.ru (M.A.A.); helgi.schioth@neuro.uu.se (H.B.S.); 2Advanced Molecular Technology, Limited Liable Company (LLC), 354340 Moscow, Russia; tarasov-v-v@mail.ru (V.V.T.); tchoubarov@mail.ru (V.N.C.)

**Keywords:** pharmacogenetics, pharmacogenomics, pain management, non-steroidal anti-inflammatory drugs, NSAID, antidepressants, safety, efficacy, genetic polymorphisms

## Abstract

Background: This systematic review summarizes the impact of pharmacogenetics on the effect and safety of non-steroidal anti-inflammatory drugs (NSAIDs) and antidepressants when used for pain treatment. Methods: A systematic literature search was performed according to the preferred reporting items for systematic reviews and meta-analysis (PRISMA) guidelines regarding the human in vivo efficacy and safety of NSAIDs and antidepressants in pain treatment that take pharmacogenetic parameters into consideration. Studies were collected from PubMed, Scopus, and Web of Science up to the cutoff date 18 October 2021. Results: Twenty-five articles out of the 6547 initially detected publications were identified. Relevant medication–gene interactions were noted for drug safety. Interactions important for pain management were detected for (1) ibuprofen/*CYP2C9*; (2) celecoxib/*CYP2C9*; (3) piroxicam/*CYP2C8*, *CYP2C9*; (4) diclofenac/*CYP2C9*, *UGT2B7*, *CYP2C8*, *ABCC2*; (5) meloxicam/*CYP2C9*; (6) aspirin/*CYP2C9*, *SLCO1B1*, and *CHST2*; (7) amitriptyline/*CYP2D6* and *CYP2C19*; (8) imipramine/*CYP2C19*; (9) nortriptyline/*CYP2C19*, *CYP2D6*, *ABCB1*; and (10) escitalopram/*HTR2C*, *CYP2C19*, and *CYP1A2*. Conclusions: Overall, a lack of well powered human in vivo studies assessing the pharmacogenetics in pain patients treated with NSAIDs or antidepressants is noted. Studies indicate a higher risk for partly severe side effects for the *CYP2C9* poor metabolizers and NSAIDs. Further in vivo studies are needed to consolidate the relevant polymorphisms in NSAID safety as well as in the efficacy of NSAIDs and antidepressants in pain management.

## 1. Introduction 

Pain is contemplated as a substantial clinical, social, and economic issue globally [1] and is the characterizing symptom for many important diseases. The International Association for the Study of Pain (IASP, www.iasp-pain.org) and the World Health Organization (WHO) define pain as “an unpleasant sensory and emotional experience associated with actual or potential tissue damage or described in terms of such damage”. Based on a neurobiological perspective, pain is divided into three types: nociceptive, inflammatory, or pathological. Dependent on the frequency, pain is furthermore divided into a chronic or acute type [2]. The number of studies that have investigated the prevalence of pain conditions are in general small and have mostly focused on low back pain [1]. There are several risk factors that are associated with pain such as age and sex. Studies in children have shown that girls experience more pain than boys. In adults, the severity, duration, and frequency of pain are higher in women compared to men [3,4]. The prevalence of any pain in adults aged 18–25 years old was investigated and found to be 66.9%. Additionally, the prevalence of chronic pain rises continuously with age. While chronic pain prevalence in adults aged 18–25 is estimated to be 14.3%, it is about 66% in the over 75 year old age group [5,6].

There are several different treatment approaches for pain. Several drug classes such as non-steroidal anti-inflammatory drugs (NSAIDs), opioids, corticosteroids, antidepressants, or anticonvulsants are used to relieve different types of pain. The interindividual variability in drug response remains a relevant clinical problem in pain treatment [7]. Furthermore, these medicines are associated with several relevant adverse events, especially during long-term usage, which may have the potential to increase morbidity and mortality. Studies have shown that long-term opioid therapy increases the risk for a diagnosis of opioid abuse or dependence [8]. Furthermore, opioid usage of at least 180 days over a 3.5 year period was associated with an increased risk for myocardial infarction [9]. NSAIDs are associated with a higher risk of gastrointestinal (GI) bleeding [10]. Additionally, fatigue, somnolence, and dizziness are common adverse effects that have been reported in patients using antidepressants [11]. 

Genetics plays a substantial role in the interindividual variability in drug response as it influences the patient’s sensitivity or resistance to drugs [12]. Pharmacogenetics is an important tool that can be used to elucidate the genetic basis for the absorption, distribution, metabolism, and excretion of drugs in different patients [13]. Various polymorphisms in genes expressing drug-metabolizing enzymes of analgesic therapeutics, receptors, and molecules in the pain pathway have been tested regarding their suitability for the prediction of the efficacy and safety of pain medications. Therapeutics used against pain are in general strongly metabolized by hepatic cytochrome P450 enzymes. Although the knowledge of single-nucleotide polymorphisms (SNPs) influencing opioid metabolism can be overall considered as still limited, studies have shown that SNPs in the genes encoding CYP450 enzymes are associated with variations in the plasma concentrations of opioids. Several opioids are metabolized by CYP2D6 such as codeine, which is metabolized to morphine. It has been found that the plasma concentration of morphine is higher in patients with ultra-rapid metabolizers (UM) compared to patients with poor metabolizers (PM). As a result, UM patients are liable to the adverse effects, while PM patients experience a poor analgesic effect [14,15]. Two recent systematic reviews have summarized the current knowledge of the impact of pharmacogenetics on opioids [16,17]. Although NSAIDs and antidepressants are important pillars in the current treatment strategies of pain, associations between the pharmacogenetically relevant SNPs and changes in the safety and efficacy of these drug groups are still controversial. 

Due to the current lack of reliable genetic markers that can be used as a single predictive tool for the treatment outcome of pain, the present treatment strategies are mainly based on clinical grounds. The aim of this review was to elucidate the pharmacogenetic evidence and highlight genes that play a role in the pharmacodynamics and safety of anti-inflammatory drugs and antidepressants used in pain treatment. Furthermore, we describe the limitations and possibilities regarding the usage of genetic markers as a tool for individual strategies in pain treatment. 

## 2. Pain Management 

Pain management is one of the most important components of the symptomatic treatment of many diseases in all areas of medicine [18]. The quality and timeliness of pain treatment not only determine the patient’s quality of life, but also the prognosis of the disease [19]. The modern concept of pain treatment as determined by the WHO is a “step concept” or “ladder concept”. The main idea of this is that treatment begins with the least active and safest drugs, while it suggests using more active drugs that also have more serious side effects if necessary [20,21]. Another fundamental principle of pain treatment is timeliness, the availability of pain relief, and the choice of an adequate method of pain relief for each patient based on the type of disease, living conditions, financial, and social capabilities [19]. It should be noted that modern pain treatment not only includes pharmacological methods, but also nonpharmacological approaches, for example, surgical and physiotherapeutic methods [21]. In our work, we focused specifically on the pharmacological methods of pain treatment. 

According to the WHO pain ladder (Figure 1) and the American Pain Society guidelines, acetaminophen should be used in the first step of pharmacological pain treatment (with a normal liver function as a prerequisite). Alternatives are aspirin (while avoiding a concomitant use of glucocorticoids) and drugs from the NSAID group (in pain conditions with a relevant of inflammatory component, or in the case of metastatic bone lesions) [20,21,22,23]. Both non-selective cyclooxygenase (COX) inhibitors (e.g., diclofenac or indomethacin) and selective cyclooxygenase 2 (COX-2) inhibitors (celecoxib) can be used depending on the individual ratio of the potential benefits to side effects, pharmacogenetics, and the type of pathology of a particular patient [24]. It should be noted that in some situations (such as neuropathic pain, fibromyalgia, or neuralgia), pain treatment should start immediately with neurotropic drugs from the antidepressant group (tricyclic antidepressants (TCAs), selective serotonin reuptake inhibitors (SSRIs), and serotonin-norepinephrine inhibitors (SNRIs)) or with anticonvulsants [25,26]. According to the European Pain Federation (EFIC), the treatment of cancer-associated pain requires an individual approach with the use of different classes of drugs depending on the pathophysiology of pain in a particular patient. In addition to the drugs described above, corticosteroids (prednisolone, dexamethasone), bisphosphonates (pamidronate), monoclonal antibodies (denosumab, tanezumab), and others (topical lidocaine, ketamine) may be used [27]. The treatment of pain in some acute or life-threatening pathologies (such as acute myocardial infarction or severe trauma) involves the use of opioid medications. Thus, the “stepwise principle“ suggested by the WHO can be varied in the latter mentioned cases [28].

The second stage of pain treatment implies a slow transition to more active drugs with more serious side effects. The main drugs used in the second step include weak opioid analgesics (such as tramadol) or strong low-dose opioid analgesics (fentanyl, tapentadol, trimepiridine, or morphine) [20,21,22,23]. Opioids should be added to the treatment gradually with the lowest effective dose and may be used in combination with first-stage medications (NSAIDs and paracetamol can potentiate the effects of opioids). The risks of adverse effects should be considered in the case of prescribing high-dose opioids or strong opioids [20,21,29].

In the third stage of pain management, high-dose strong opioid analgesics (morphine, oxycodone, fentanyl, or tapentadol) or combinations thereof are used [20,21,22,23]. In some cases, opioids and opioid receptor blockers (naloxone, naltrexone) may be used together to reduce the risk of addiction and other adverse effects [20,21,30].

## 3. Methods

### 3.1. Search Strategy

This review was registered in the open science framework (OFS), DOI number: 10.17605/OSF.IO/N9VZ3 and the protocol can be accessed via the following link: https://doi.org/10.17605/OSF.IO/N9VZ3, https://archive.org/details/osf-registrations-n9vz3-v1 (accessed on 5 December 2021). Following the PRISMA guidelines, PubMed, Scopus, and Web of Science databases were used to search for peer-reviewed publications using the following search terms in different combinations and different long combination chains (“pharmacogenetics” OR “pharmacogenomics”) AND (“anti-inflammatory” OR “Aspirin” OR “Cox inhibitor” OR “NSAID” OR “corticosteroid” OR “tricyclic antidepressant, TCA” OR “selective serotonin reuptake inhibitors, SSRI” OR “serotonin and norepinephrine reuptake inhibitors, SNRI” OR “selective norepinephrine reuptake inhibitors, selective noradrenaline reuptake inhibitors, NRI” OR “tetracyclic antidepressant” OR “Monoamine oxidase inhibitors, MAOI”) AND (“Cytochrome P450” OR “UDP Glucuronosyltransferase” OR “UGT” OR “CYP2C9” OR “CYP2C8” OR “CYP2C19” OR “CYP2B6” OR “CYP2D6” OR “CYP1A2” OR “CYP3A4” OR “CYP3A5” OR “COMT” OR “ABCB1” OR “SLC6A4, SERT” OR “NET, SLC6A2” OR “BDNF” OR “5-hyroxytryptamine receptor, 5-HT receptor, HTR, 5-HT” OR “Melatonin receptor, MT”). To catch an as wide as possible range of papers, the initial protocol was amended by adding the additional search terms “Genetic polymorphism” OR “Safety” OR “Efficacy“ AND/OR the drug names listed in Table 1. All studies with a publication date until 18 October 2021 were included. Duplicates, book chapters, editorials, meeting abstracts, and proceedings papers were excluded. Articles were included if they compared the efficacy or safety of a drug by considering the pharmacogenetics in the area of pain including human subjects. 

### 3.2. Study Selection

Farzin Zobdeh, Ivan Eremenko, and Mikail Akan performed a double screening with the references obtained during the database searches. The first screening included only the information available in the titles and abstracts. A study had to (1) be written in English, (2) mention pharmacological pain treatment and pharmacogenetics, and (3) represent an original article to be considered in our review. In a further screening step, at least one reviewer needed to confirm the reference. The first screening was performed using abstrackr (http://abstrackr.cebm.brown.edu/account/login (accessed on 12 December 2021)).

In the case that the title and abstract were not sufficient to determine whether a study needed to be excluded, a second screening step was performed. For the second screening step, the full text of the study was considered including any supplementary material available. To be included here, the studies had to meet the following inclusion criteria: (1) Describe the interplay between the pharmacological pain treatment and genetic response (therapy efficacy or safety) highlighting the main genes and pathways involved, and (2) represent human studies (i.e., animal studies were excluded). 

### 3.3. Data Extraction and Synthesis

A Microsoft Excel spreadsheet file was generated to include all of the studies selected during the first screening step. From the papers selected for the systematic review, the following data were extracted: First author; Title; Journal; Year of publication; Country of origin.

On another spreadsheet, the following information was extracted from the studies selected during the second screening: The same information from the first Excel spreadsheet; Ethnicity of participants; Number of participants; Percentage of females; Outcome measured; Gene assessed; Variants; Major findings of the study.

A search of the PubMed, Scopus, and Web of Science databases identified a total of 6547 citations. Publications were then selected according to our criteria above-mentioned. Eventually, we selected 25 studies to be presented in this review (Figure 2) Of note, no studies matching the selection criteria were found for the drug class corticosteroids, and a search on distinctive compounds belonging to this drug class was not further refined to keep the focus on NSAIDs and antidepressants. 

## 4. Drug Groups

This review identified 25 articles that assessed 11 drugs from three major drug groups used for pain treatment (i.e., NSAIDs, TCAs, SSRIs that fulfilled the criteria to be included in the evaluation regarding the impact of genetic variation on pain reduction efficacy and drug safety). These articles are discussed in the following sections. 

### 4.1. Nonsteroidal Anti-Inflammatory Drugs and Aspirin

NSAIDs are widely used in multiple medical conditions and are an important pillar in the treatment of pain. NSAIDs primarily inhibit COX-1 and COX-2, leading to analgesic, antipyretic, and anti-inflammatory effects [31]. 

Various enzymes are involved in the metabolism of NSAIDs, most importantly, members of the cytochrome P450 enzyme family such as CYP2C9, CYP2C8, and CYP3A4 [32,33]. Besides the CYP family, UDP-glucuronosyltransferases (UGTs) such as UGT2B7, UGT1A6, UGT1A4, and UGT1A9 also contribute to NSAID metabolism. The mentioned proteins show a various degree of variability in activity due to SNPs, which may affect individual NSAID responses (Table 2).

#### 4.1.1. Ibuprofen

Ibuprofen is a NSAID, which is widely used for pain treatment. Ibuprofen enfolds its analgesic effect via the inhibition of COX-1 and COX-2, which leads to the inhibition of prostaglandin formation [34]. Moreover, a racemic mixture of *S*-(+) and *R*-(−) enantiomers, with *S*-ibuprofen being pharmacologically active and the *S*- and *R*-enantiomers being capable of converting into *S*-ibuprofen. Both *S*- and *R*-enantiomers are mostly metabolized by CYP2C9, whereas CYP2C8 contributes, to a lesser extent, to the metabolism of the *R*-enantiomer [35]. 

In a study investigating ibuprofen pharmacokinetics in 122 healthy volunteers, Ochoa et al. [36] showed that the *CYP2C9* variants *CYP2C9*2* and **3* led to significantly lower *S*-ibuprofen clearance and a higher *S*-ibuprofen plasma concentration compared to the wild-type carriers. Despite this, there were no differences seen in the *R*-ibuprofen pharmacokinetics in dependence of the *CYP2C9* genotype. The *CYP2C8* polymorphisms had no significant effect on the *S*-ibuprofen or on *R*-ibuprofen pharmacokinetics. Of note, gender seems to have an impact on the ibuprofen pharmacokinetics as shown with a lower *R*-ibuprofen half-life and *S*-ibuprofen plasma concentration in women carrying the wild-type of *CYP2C9* and *CYP2C8* [37]. However, the functional importance of the pharmacokinetic differences between the *CYP2C9* phenotypes in ibuprofen pain reduction, discussed above, remains rather unclear in terms of therapy efficacy.

**Efficacy.** In a randomized controlled trial assessing the ibuprofen analgesic efficacy after dental surgery in 43 patients conducted by Saiz-Rodríguez et al. [37], a trend toward a greater pain score reduction 6 h after ibuprofen intake was observed in the *CYP2C9* PMs compared with the intermediate (IM) or normal (NM) metabolizers was observed. However, no statistically significant impact of the *CYP2C9* and *CYP2C8* phenotypes in the pain reduction measures was found, which is in concordance with the data observed by Weckwerth et al. on 200 patients after lower third molar extraction [38]. In addition, Jaja et al. [39] indicated that emergency department visits due to severe pain episodes in 165 sickle cell disease patients receiving NSAIDs including 50 patients receiving ibuprofen were related to the *CYP2C9* phenotypes. The study showed that NMs visited the hospital more often than IMs, while no difference in emergency department visits was observed in dependence of the *CYP2C8* phenotypes. In summary, it appears that genetic variation in *CYP2C9* and *CYP2C8* only have a very modest influence on the ibuprofen analgesic efficacy (Table 2).

**Safety.** Ochoa et al. studied the effect of polymorphisms in *CYP2C9* and *CYP2C8* and of gender on the pharmacokinetics of ibuprofen as above-mentioned. In the framework of the study, the authors reported multiple adverse effects including hepatic profile alteration, acute rhabdomyolysis, aural skin rash, headache, or abdominal pain in seven out of 122 volunteers. However, no association with gender, *CYP2C9*, or *CYP2C8* genotype was detected [36]. Martínez et al. reported that the inherited impairment of *CYP2C9* activity increased the risk of acute GI bleeding, as studied in 94 patients treated with different types of NSAIDs including celecoxib, ibuprofen, diclofenac, and indomethacin (Table 2) [40]. The Clinical Pharmacogenetics Implementation Consortium (CPIC) defines an activity score ranging from 0 to 2 based on the *CYP2C9* diplotype status. According to this system, subjects with a score 0 to 0.5 are classified as PMs, individuals with a score between 1 to 1.5 are IMs, and those with a score of 2 are NMs. CPIC recommends starting ibuprofen with the lowest recommended dose accompanied by the monitoring of adverse effects in the case of IMs (activity score 1) or with 25–50% of the lowest recommended dose in the case of a *CYP2C9* PM status to avoid side effects [41].

#### 4.1.2. Celecoxib

Celecoxib is used for the treatment of rheumatoid arthritis and osteoarthritis and exhibits anti-inflammatory, antipyretic, and analgesic properties by inhibiting COX-2 [42]. Celecoxib is primarily metabolized by CYP2C9 and, to a lesser extent, by CYP3A4 [43], further oxidized by cytosolic alcohol dehydrogenases ADH1 and ADH2 [44], and conjugated by UGTs [45].

When investigating the impact of *CYP2C9* polymorphisms on celecoxib pharmacokinetics in 21 healthy volunteers, Kirchheiner et al. [46] detected that *CYP2C9*3* allele carriers showed a reduced oral clearance of celecoxib and elimination half-life compared to the *CYP2C9*1/*1* genotype carriers. However, conflicting data were obtained by Brenner et al. [47], showing no correlation between the celecoxib pharmacokinetics and the *CYP2C9* genotype in 12 healthy participants. Of note, more recent studies have shown a significant impact of *CYP2C9* polymorphisms on the celecoxib pharmacokinetics, as demonstrated in 39 healthy volunteers [48]. Nonetheless, the association of these findings with the pharmacodynamic outcome (i.e., pain relief efficacy) is rather uncertain to date and needs further investigation.

**Efficacy.** We identified three studies investigating the impact of pharmacogenetics on celecoxib efficacy. Hamilton et al. [49] studied the postoperative pain management following total knee arthroplasty in 31 patients. Here, no conclusions about an association between the *CYP2C9* genotype and celecoxib efficacy in pain reduction could be drawn due to the multiple concomitant drug administration. In a randomized controlled trial of oral celecoxib administration in 195 children after adenotonsillectomy performed by Murto et al. [50], it was shown in 93 genotyped participants that celecoxib led to reduced pain recurrence in *CYP2C9*3* allele carriers in comparison to the subjects carrying the wild-type allele. Although a slight difference was obtained, these data are insufficient to make clear conclusions on the role of the *CYP2C9* genotype in celecoxib analgesic efficacy and further studies are needed [36,40]. Ustare et al. reported that the response to celecoxib was better for postoperative pain in two patients among 99 IM patients with the *CYP2C9*1/CYP2C9*3* genotype compared to the UM and PM patients (Table 2) [51].

**Safety.** Murto et al. did not detect any association between the *CYP2C9* genotype and the frequency changes in adverse effects in their study described above [50]. No additional human in vivo studies exploring the impact of pharmacogenetics on celecoxib safety were identified. Of note, in a case report, Gupta et al. reported a relation between the intermediate *CYP2C9* metabolizers and GI bleeding in a 24-year-old female patient with constant lower abdominal and pelvic pain who developed severe gastropathy after celecoxib usage (Table 2) [52]. The CPIC recommends initiating the therapy with the lowest recommended dose and increasing the dose in a stepwise fashion and monitoring individuals for side effects who are *CYP2C9* IMs. For individuals who are *CYP2C9* PMs, a 50–75% reduction in the initial dose and a cautious dose titration is recommended [41].

#### 4.1.3. Piroxicam

Piroxicam is a NSAID of the oxicam class, which is used to relieve the symptoms of painful inflammatory conditions such as rheumatoid arthritis by mainly inhibiting COX-1. Like other NSAIDs, piroxicam is mostly metabolized by CYP2C9 and, to a lesser extent, by CYP2C8 [41,53]. Perini et al. detected an impaired oral clearance and increases in the inhibition of COX-1 activity in individuals carrying *CYP2C9*1/*2* or *CYP2C9*1/*3* compared to the wild-type carriers [53].

**Efficacy.** Calvo et al. investigated the *CYP2C9* and *CYP2C8* genotypes in relation to piroxicam efficacy after dental surgery in 102 patients [54]. The study detected no differences between the wild-type patients and mutant allele carriers. This finding suggests that there is no influence of the *CYP2C9* and *CYP2C8* genetic variation on piroxicam response. However, due to a limited number of studies available, this relationship should be further investigated in future trials (Table 2).

**Safety.***CYP2C8* and *CYP2C9* polymorphisms have been repeatedly described to be associated with an increased number of adverse reactions of NSAIDs. Calvo et al. [54], who studied the relation between *CYP2C8* and *CYP2C9* genotypes and the clinical efficacy of oral piroxicam as above-mentioned, detected two subjects among the 102 individuals with adverse reactions who both carried the *CYP2C8*3* mutant and the *CYP2C9*1/*3* genotype. Subjects carrying the *CYP2C8*3* mutant and the *CYP2C9*1/*3* genotype reported sleepiness and stomachaches [54]. According to the data that we collected, no other human in vivo study was identified to have investigated piroxicam safety in relation to the pharmacogenetic SNPs in a systematic manner (Table 2). The CPIC recommends considering alternative therapy approaches in individuals who are *CYP2C9* IMs and PMs [41].

#### 4.1.4. Dexketoprofen

Ketoprofen belongs to the propionic acid class of NSAIDs and has antipyretic, anti-inflammatory, and analgesic effects by non-selective COX inhibition. The *S*-(+) enantiomer, dexketoprofen, is pharmacologically active, while the *R*-(−) enantiomer is inactive [55,56]. Ketoprofen is primarily metabolized by UGTs such as UGT2B7, UGT2B4, and UGT1A3 [55]. It is also metabolized by CYP2C9, although to a lesser extent [57].

Mejía-Abril et al. [58] investigated multiple variants in various genes encoding transporters and metabolizing enzymes including *CYP1A2*, *CYP2A6*, *CYP2B6*, *CYP2C8*, *CYP2C9*, *CYP2C19*, *CYP2D6*, *CYP3A4*, *CYP3A5*, *UGT1A1*, *ABCB1*, *ABCC2*, *SLCO1B1*, and *SLC22A1* and their impact on dexketoprofen pharmacokinetics in 85 healthy volunteers. No significant association between any of the assessed genes and dexketoprofen pharmacokinetics has been observed. The obtained data may be explained by the little involvement of CYP isozymes in ketoprofen metabolism. Moreover, the assessed UGT1A1 is not the main enzyme in ketoprofen glucuronidation [59], which may also be the reason for the observed results.

**Efficacy.** To date, no in vivo study has been found that evaluated the genetic variations in enzymes involved in ketoprofen metabolism and their effect on pain reduction (Table 2).

**Safety.** The most common adverse effects of dexketoprofen are GI side effects including nausea, vomiting, abdominal pain, flatulence, constipation, dyspepsia, diarrhea, and others that are less frequent [55]. Only one study has been identified in the literature that addressed dexketoprofen safety in relation to pharmacogenetics. Mejía-Abril et al. [58] reported no association between the genetic polymorphisms of *CYP1A2*, *CYP2A6*, *CYP2B6*, *CYP2C8*, *CYP2C9*, *CYP2C19*, *CYP2D6*, *CYP3A4*, *CYP3A5*, *UGT1A1*, *ABCB1*, *ABCC2*, *SLCO1B1*, and *SLC22A1* and the adverse effects of dexketoprofen. This study included 85 healthy volunteers enrolled in three clinical trials (Table 2) [58].

#### 4.1.5. Diclofenac

Diclofenac sodium is a drug used in painful and rheumatoid conditions. Diclofenac acts as a non-selective COX inhibitor and is mainly metabolized by *CYP2C9* and, to a lesser extent, by *CYP2C8* and *CYP3A4* [60,61,62]. 

It was shown that some link exists between the *CYP2C9* genotype and diclofenac pharmacokinetics. Two studies performed on 102 and 160 healthy volunteers detected a decrease in diclofenac metabolism in participants carrying the *CYP2C9*3* allele [63,64]. Moreover, oxidative metabolism of diclofenac is performed by CYP2C9 through 4-hydroxylation. Aithal et al. revealed only slightly lower 4-hydroxylation rates of *CYP2C9*3* carriers among 24 patients [65]. However, conflicting data are available, which proposed no association between the *CYP2C9* genotype and diclofenac pharmacokinetics in 12 healthy participants [48]. 

**Efficacy.** No studies have been identified that have evaluated the relationship between pain treatment response and pharmacogenetic parameters. Thus, to date, it remains unknown as to what extent the genetic variation influences the diclofenac treatment outcome (Table 2).

**Safety.** Diclofenac is one of the most common drugs causing idiosyncratic hepatotoxicity with a considerable rate of severe events [66,67]. Daly et al. [68] studied the association between the genetic predisposition and diclofenac-induced hepatotoxicity in 24 patients who had suffered from diclofenac hepatotoxicity. The authors detected allelic variants of *UGT2B7* (*UGT2B7*2*), *CYP2C8* (*CYP2C8*4*), and *ABCC2* (*ABCC2 C-24T*) to be related to diclofenac hepatotoxicity by forming and accumulating reactive diclofenac metabolites (Table 2) [68].

#### 4.1.6. Meloxicam

Meloxicam is an NSAID indicated for the treatment of osteoarthritis and rheumatoid arthritis. Meloxicam acts as a selective COX-2 inhibitor and is metabolized mostly by CYP2C9 and, to a lesser extent, by CYP3A4 [69].

Investigating the association of meloxicam pharmacokinetics with *CYP2C9* genotypes in 22 healthy volunteers, Lee et al. [70] showed that *CYP2C9*3* allele carriers had a significantly decreased meloxicam metabolism compared to the *CYP2C9*1/*1* carriers. These data are supported by more recent studies [71,72]. 

**Efficacy.** No in vivo studies have been reported testing the impact of genetic variants on meloxicam efficacy in pain treatment. The observed difference in meloxicam pharmacokinetics in dependence of the *CYP2C9* genotype may hint at interindividual variability in meloxicam response, thus this association should be further investigated in the setting of pain treatment (Table 2).

**Safety.** Like other NSAIDs, meloxicam may cause several side effects including GI adverse effects (nausea, vomiting, inflammation, or ulceration) and rarely, myocardial infarction or stroke. Lee et al. detected a correlation between *CYP2C9*3/*3* and the risk for adverse effects such as GI bleeding or cardiovascular events. This study was intended to evaluate the pharmacokinetic relation between meloxicam and the genetic polymorphisms as above-mentioned (Table 2) [70]. The CPIC recommends for meloxicam treatment of *CYP2C9* IMs with an activity score of 1 to reduce the lowest recommended starting dose by 50% and to carefully increase the dose until a steady state is reached. For patients who are *CYP2C9* PMs, alternative therapy approaches should be considered [41].

#### 4.1.7. Aspirin

Acetylsalicylic acid, or aspirin, has analgesic, antipyretic, and anti-inflammatory properties and is used for the treatment of rheumatoid arthritis and the prevention of cardiovascular diseases. Aspirin acts as a non-selective COX inhibitor and is mainly metabolized by UGTs such as UGT1A6, UGT1A1, UGT1A7, and UGT1A9 [73,74,75]. 

Chen et al. [76] assessed the relationship between aspirin salivary and urinary pharmacokinetics and the *UGT1A6* genotype in 25 healthy volunteers and discovered that *UGT1A6*2/*2* genotype carriers showed more rapid glucuronidation of aspirin. In a further investigation of this relationship, van Oijen et al. [77] measured the plasma concentrations of aspirin in 60 healthy participants and detected a significantly faster aspirin metabolism in the *UGT1A6*2/*2* genotype carriers. 

**Efficacy.** There is evidence for a strong relationship between the *UGT1A6* genotype and aspirin pharmacokinetics, however, no studies have been found to have evaluated the clinical implication of this relationship to date, thus its effect on individual aspirin response in pain conditions remains to be determined (Table 2).

**Safety.** Long-term usage of aspirin raises concerns regarding the risk of GI bleeding [78]. Shiotani et al. [79] detected that carriers of the *SLCO1B1*1b* haplotype and the *CHST2 2082 T* allele were at significantly higher risk for peptic ulcer and ulcer bleeding compared to the controls when using aspirin. This study consisted of three groups, patients with peptic ulcer (*n* = 111), patients with GI bleeding “(*n* = 45), and a control group (*n* = 482), and aimed to investigate the relationship between the pharmacogenetics and low-dose aspirin-induced GI bleeding and peptic ulcer [79]. Moreover, Figueiras et al. revealed that patients who consumed a mean defined daily dose of NSAIDs greater than 0.5 and who carried the *CYP2C9*3* allele had a significant increase of risk for upper gastrointestinal bleeding (UGIB) compared with patients who took the same dose but were non carriers of this variant (Table 2) [80]. Wang et al. investigated the association between genetic variants of tumor necrosis factor α (TNF-α) and UGIB [81]. This study included 154 patients with coronary artery diseases who took low-dose aspirin. This study tested TNF-α gene polymorphisms (including three SNPs; *TNF-α -1031T > C*, *TNF-α -863C > A*, and *TNF-α -857C > T*). The group reported that C allele carriers of *TNF-α -1031T > C* and A allele carriers of *TNF-α -863C > A* had a significantly increased risk of aspirin-induced UGIB compared to patients carrying the C or T allele of *TNF-α -857C > T*, which was not associated with UGIB [81]. The vitamin K epoxide reductase (VKOR) enzyme regulates the vitamin K level. Groza et al. [82] tested the possible correlation between the *VKORC1 -1639 G > A* polymorphism and UGIB among 163 patients diagnosed with UGIB. This study showed that subjects with NSAID- or aspirin-induced non-variceal UGIB were significantly more often carrying the *VKORC1 -1639 G > A AA* genotype compared to the control group without UGIB [82]. Piazuelo et al. examined the association between UGIB and nitric oxide synthase (*eNOS*) (*a* and *b* allele) and the platelet glycoprotein (*GPIIIa*) (*PIA1* and *PIA2* allele) genes [83]. This study included 88 patients with UGIB who used low-dose aspirin, 108 control subjects who were low-dose aspirin users and did not have UGIB, and 158 blood-donors as a second control group. This study did not show an association between the *GPIIIa PlA1/A2* polymorphism and UGIB. However, a significantly lower UGIB risk was associated with the *a* allele of *eNOS* in patients taking low-dose aspirin [83].

**Table 2 pharmaceutics-14-01190-t002:** Studies investigating the association between pharmacogenetics and the effect and safety of NSAIDs and aspirin in pain in vivo (human studies).

Study	Drug	Ethnicity	Study Design	Outcome	Gene Assessed	Variants	Findings (Effect or Safety)
Martinez et al. (2004) [40]	Celecoxib, diclofenac, ibuprofen, piroxicam	N/A	Patients with GI bleeding (*n* = 94) and healthy individuals (*n* = 124)	Adverse effects of different NSAIDs	*CYP2C9*	*CYP2C9*1/*1*, **1/*2*, **1/*3*, **2/*2*, **2/*3*, **3/*3*	*CYP2C9*2**allele* frequency increased in patients with acute bleeding
Saiz-Rodriguez et al. (2021) [37]	Ibuprofen	White	43 patients with moderate to severe pain after dental surgery	Ibuprofen response	*CYP2B6 CYP2C8 CYP2C9 CYP2C19 CYP2D6 CYP3A4 PTGS2*	*CYP2B6 G/G*, *G/T*, *T/T* *CYP2C8* PMs, IMs, and NMs *CYP2C9* PMs and IMs *CYP2C19* IMs, NMs, and UMs *CYP2D6* PMs, IMs, NMs, and UMs	Greater pain reduction 6 h after ibuprofen intake in *CYP2C9* PMs compared with IMs/NMs
Weckwerth et al. (2020) [38]	Ibuprofen	Brazil	200 patients with acute pain	Ibuprofen response	*CYP2C8 CYP2C9*	*CYP2C8*1/*1*, **1/*2*, **1/*3*, **1/*4*, **2/*3*, **3/*4* *CYP2C9*1/*1*, **1/*2*, **1/*3*, **2/*2*, **2/*3*, **3/*3*	*CYP2C9* and *CYP2C8* IMs and PMs have lower levels of postoperative pain
Jaja et al. (2015) [39]	Ibuprofen, aspirin	African American	50 patients with sickle cell disease	NSAIDs efficacy	*CYP2C8*, *CYP2C9*	*CYP2C8*1/*1*, **1/*2*, **1/*3*, **1/*4*, **2/*2*, **2/*3*, **2/*4* *CYP2C9*1/*1*, **1/*2*, **1/*3*, **1/*5*, **1/*6*, **1/*8*, **1/*9*, **1/*11*, **2/*3*, **5/*9*, **6/*8*, **8/*9*, **9/*11*	*CYP2C9* NMs visited the hospital more frequently due to severe pain
Hamilton et al. (2020) [49]	Celecoxib	N/A	31 patients with postoperative pain	Celecoxib efficacy and safety	*CYP2C9*	*CYP2C9* NMs and IMs	Concomitant drug intake, no clear conclusion regarding a pharmacogenetic association
Murto et al. (2015) [50]	Celecoxib	Caucasian, African American Hispanic, South and East Asian	93 patients with postoperative pain	Celecoxib efficacy	*CYP2C9*	*CYP2C9*1/*1*, **1/*2*, **1/*3*, **2/*3*, **2/*2*, **3/*3*	Reduced pain recurrence in *CYP2C9*3* allele carriers compared to wild-type carriers
Ustare et al. (2020) [51]	Celecoxib	Malay, Malay-Chinese, Malay-Polynesian, Filipinos	99 patients with postoperative pain	Celecoxib efficacy	*CYP2C9*	*CYP2C9*1/*1*, **1/*3*	Lower pain scores in *CYP2C9* IMs after 24 and 48 h compared to NMs
Calvo et al. (2017) [54]	Piroxicam	Brazil	102 patients with postoperative pain	Piroxicam efficacy and adverse effects	*CYP2C8*, *CYP2C9*	*CYP2C8*1/*1*, **1/*3*, **3/*3* *CYP2C9*1/*1*, **1/*2*, **1/*3*, **2/*2*, **2/*3*, **3/*3*	Postoperative pain scores and adverse effects were comparable between genotypes
Daly et al. (2007) [68]	Diclofenac	North European	Patients with and without diclofenac-induced hepatotoxicity (*n* = 28/48) Healthy volunteers (*n* = 112)	Diclofenac adverse effects	*UGT2B7*, *CYP2C8*, *ABCC2*	*UGT2B7*1/*1*, **1/*2*, **2/*2* *CYP2C8*1/*1*, **1/*2*, **1/*3*, **1/*4*, *1/*5*, **2/*3*, **2/*2* *ABCC2 C-24/C-24*, *C-24/T-24*, *T-24/T-24*	*UGT2B7*2* allele was associated with a higher risk of diclofenac-induced hepatotoxicity compared with wild-type carriers
Aithal et al. (2000) [65]	Diclofenac	Caucasian	124 patients with diclofenac-induced hepatotoxicity (*n* = 24); control group (*n* = 100)	Diclofenac adverse effects	*CYP2C9*	*CYP2C9*1/*1*, **1/*2*, **1/*3*, **2/*3*, **3/*3*	No association of *CYP2C9*2* or *CYP2C9*3* with diclofenac-induced hepatotoxicity
Shiotani et al. (2014) [79]	Aspirin	Japanese	638 patients with peptic ulcer (*n* = 111); patients with GI bleeding (*n* = 45); control group (*n* = 482)	Aspirin adverse effects	*SLCO1B1*, *CHST2*	*SLCO1B1 388 A > G (rs2306283)*, *521 T > C (rs4149056)* *CHST2 2082 C > T (rs6664)*	*SLCO1B1*1b* and *CHST2 2082 T* allele frequency was increased in patients with peptic ulcer and ulcer bleeding compared to the controls
Wang et al. (2019) [81]	Aspirin	N/A	154 patients with coronary heart disease; with (*n* = 57) or without (*n* = 97) upper GI bleeding (UGIB)	Aspirin adverse effects	*TNF-α* gene	*-1031T > C TT*, *TC*, *CC* *-863C > A CC*, *CA*, *AA* *-857C > T CC*, *CT*, *TT*	*-1031T > C*: *C* allele and *CC* genotype carriers and -*863C > A*: *A* allele, *CA*, and *CA + AA* genotype carriers had increased risk of UGIB *-857 C > T* had no effect
Groza et al. (2017) [82]	Aspirin	N/A	Patients with UGIB (*n* = 154); control group (*n* = 178)	Aspirin adverse effects	*VKORC1*	*VKORC1 -1639 G > A GG*, *GA*, *AA*	*VKORC1 -1639 G > A*: *AA* genotype is associated with an increased risk of UGIB
E. Piazuelo et al. (2008) [83]	Aspirin	White	Patients with UGIB (*n* = 88); control patients (*n* = 108)	Aspirin adverse effects	*eNOS* *GP IIIa*	*eNOS 4b/4b*, *4a/4b*, *4a/4a* *GP IIIa PlA1/A1*, *PlA1/A2*, *PlA2/A1*	*eNOS a* allele carriers had reduced risk of UGIB
Figueiras et al. (2016) [80]	Multiple drugs	Caucasian	1920 patients with hematemesis, melena; and hematochezia (*n* = 577); control group (*n* = 1343)	NSAIDs adverse effects	*CYP2C9*	*CYP2C9*1/*1*, **1/*2*, **1/*3*, **2/*2*, **2/*3*, **3/*3*	Higher risk of upper GI bleeding in *CYP2C9*3* allele carriers *CYP2C9*2* allele had no such effect
Lee et al. (2014) [70]	Meloxicam	Korean	22 healthy participants	Meloxicam adverse effects	*CYP2C9*	*CYP2C9*1/*1*, **1/*3*, **3/*3*	*CYP2C9*3/*3* carriers have significantly greater TXB2 inhibition compared with *CYP2C9*1/*1* and **1/3* (possible differences in the incidence of cardiovascular complications and bleeding)
Mejía-Abril et al. (2021) [58]	Dexketoprofen	Caucasian, Latin-American, Black, Asian	85 healthy participants	Dexketoprofen adverse effects	*CYP1A2*, *CYP2A6*, *CYP2B6*, *CYP2C8*, *CYP2C9*, *CYP2D6*, *CYP3A4*, *CYP3A5 ABCB1*, *ABCC2*, *SLCO1B*, *UGT1A1*	*CYP1A2*1C*, **1F*, **1B* *CYP2A6*9* *CYP2B6*9*, **5*, *rs4803419*, *rs2279345*, *rs2279343* *CYP2C8*2*, **3*, **4* *CYP2C9*2*, **3* *CYP2C19*2*, **3*, **4*, **17* *CYP2D6*3*, **4*, **6*, **7*, **8*, **9*, **10*, **14*, **17*, **41* *CYP3A4*22*, *rs55785340*, *rs4646438* *CYP3A5*3*, **6* *ABCB1 C3435T*, *G2677 T/A*, *C1236T* *ABCC2 rs2273697*, *rs717620* *SLCO1B1*1B*, **5*, *rs4149015*, *rs11045879* *SLC22A1*2*, **3*, **5* *UGT1A1*80*	No adverse effects after dexketoprofen intake were reported

N/A = not available, GI = gastrointestinal, NSAIDs = non-steroidal anti-inflammatory drugs, PM = poor metabolizer, IM = intermediate metabolizer, NM = normal metabolizer, UM = ultra-metabolizer, UGIB = upper gastrointestinal bleeding, TXB2 = Thromboxane B2.

### 4.2. Antidepressants

Antidepressants are used extensively for pain treatment. A large-scale survey has indicated that antidepressants represent 3% of all the analgesic prescriptions used to treat chronic pain [84]. Moreover, antidepressants are effective in the treatment of musculoskeletal pain in fibromyalgia [85]. Regarding the relation between the efficacy and safety of antidepressants with pharmacogenetics, we were able to detect studies for the drug classes TCA and SSRI. In contrast, no clinical studies were identified for SNRIs, monoamine oxidase inhibitors (MAOIs), and tetracyclic antidepressants (Table 3).

#### 4.2.1. Tricyclic Antidepressants 

TCAs are on one hand, used to treat diseases such as depression and obsessive-compulsive disorder, while on the other hand, TCAs are also used to treat neuropathic pain. In fact, the use of TCAs in psychiatric disorders has declined and they are now more often used in neuropathic pain treatment such as diabetic neuropathy [86,87]. TCAs are acting as serotonin and norepinephrine reuptake inhibitors and are mainly metabolized by CYP2D6 and CYP2C19 [88]. An association between *CYP2D6* and *CYP2C19* phenotypes and adverse effects secondary to TCA intake has been repeatedly described for patients treated for depression. However, little data exist to evaluate the drug regarding efficacy and safety in pain patients [89].

##### Amitriptyline

Amitriptyline is indicated for the treatment of major depressive disorder (European Medicines Agency (EMA), www.ema.europa.eu; and Food and Drug Administration (FDA), www.fda.gov) and neuropathic pain (EMA). The compound is mainly metabolized by CYP2C19, CYP3A4, and CYP2D6, and to a lesser extent, by CYP1A2 and CYP2C9 [90]. 

The relationships between the *CYP2C19* and *CYP2D6* genotypes and amitriptyline pharmacokinetics have been well-studied [89]. Shimoda et al. revealed that the genotype of *CYP2C19* is one of the essential factors that influence the plasma concentrations of amitriptyline and the capacity to demethylate amitriptyline [91]. Ryu et al. [88] performed a randomized controlled trial including 24 healthy adults with the aim to study the pharmacokinetics of amitriptyline in relation to the genotypes of *CYP2C19 (CYP2C19*2/*2*, **2/*3*, *or *3/*3)* and *CYP2D6 (CYP2D6*10/*10).* This study indicated that the metabolic pathway of amitriptyline is influenced by *CYP2C19* rather than *CYP2D6.* However, this study did not detect any relation between the amitriptyline pharmacodynamics and *CYP2C19* or *CYP2D6* [88]. Additionally, Matthaei et al. reported that a decrease in the activity of CYP2D6 led to an increase in the amitriptyline plasma concentration [92]. However, there were only a limited number of studies investigating the impact of genetic variation on the efficacy of amitriptyline in neuropathic pain reduction. 

**Efficacy.** A significantly lower pain intensity level was observed in *CYP2D6* PMs compared to *CYP2D6* rapid metabolizers (RMs) during the first week of treatment of postamputation pain in 30 patients initially receiving amitriptyline [93]. 

In addition, when investigating amitriptyline response in 31 patients suffering from diabetic peripheral neuropathy, Chaudhry et al. [94] observed no significant difference in the amitriptyline analgesic efficacy in *CYP2D6* NMs by comparing them with IMs. According to the above-mentioned studies, it appears that the *CYP2D6* phenotype may have an impact on the amitriptyline treatment response in neuropathic pain conditions, although given the small sample sizes, this impact should be researched further (Table 3).

**Safety.** Chaudhry et al. [94] reported a trend toward more severe adverse effects in patients with diabetic peripheral neuropathy with lower *CYP2D6* activity scores. However, the study had a limited sample size and did not include *CYP2D6* PMs, but only IMs, NMs, and UMs. Thus, further investigation is needed to replicate this finding [94]. Steimer et al. [95] identified that patients carrying two functional *CYP2D6* alleles combined with only one functional *CYP2C19* allele showed a lower risk of side effects compared to the carriers of other combinations of alleles. This is especially noticeable for those patients with only one functional *CYP2D6* allele, as studied in 50 Caucasians with depressive disorder [95]. The study by Ryu et al., as above-mentioned, also investigated a possible association of several pharmacogenetically important SNPs with the likelihood for the anticholinergic side effects or orthostatic events with amitriptyline, but did not detect any association (Table 3) [88]. CPIC recommends a dosing according to *CYP2D6* and *CYP2C19* phenotypes based on data collected thus far in patients with depression. This includes an increase in the target therapeutic dose in *CYP2D6* UMs, a 25% reduction in the recommended starting dose in *CYP2D6* IMs, and a 50% reduction in the recommended starting dose in *CYP2D6* PMs and *CYP2C19* PMs [89]. The Dutch Pharmacogenetics Working Group (DPWG) recommends using increased doses of amitriptyline in *CYP2D6* UMs because of the higher metabolic rate of this drug, whereas *CYP2D6* IMs and PMs should receive 75% and 70% of the standard dose, respectively [96].

##### Nortriptyline 

Nortriptyline is approved by the FDA for used in depression treatment. Although not labeled for the treatment of pain by EMA or FDA, nortriptyline is used off-label for neuropathic pain, postherpetic neuralgia, and chronic pain [97]. Nortriptyline is an active metabolite of amitriptyline, which is generated via demethylation through *CYP2C19.* Nortriptyline is hydroxylated by *CYP2D6* to 10-hydroxynortriptyline, which is an inactive metabolite [95]. Several studies have indicated a possible impact of pharmacogenetically relevant SNPs on the pharmacokinetics of the drug. Matthaei et al. [92] investigated a possible association between organic cation transporter 1 (OCT1, *SLC22A1*) polymorphisms and the pharmacokinetics of amitriptyline and nortriptyline. This study detected a two times higher the time of the maximum plasma concentration (T_max_) in the volunteers with two active *OCT1* alleles (*OCT1*1*) compared to those who were carriers of one active allele (*OCT1*1*) and one inactive or reduced activity allele (*OCT1*2*, *3**, *4**) and subjects who carried two inactive or reduced activity alleles (*OCT1*2*, **3*, **4*, **5*). However, this high nortriptyline concentration could be due to one subject who had low *CYP2D6* activity and ultra-high *CYP2C19* activity.

**Efficacy.** Benavides et al. [98] examined the association between genetic markers and the analgesic effect of the combination therapy with morphine and nortriptyline or the respective monotherapies in patients with neuropathic pain. Thirty-four SNPs in genes such as *OPRM1*, *COMT*, *HT2RA*, *CYP2C19*, and *CYP2D6* were tested among the 25 neuropathic pain patients. Only the C allele of *ABCB1 rs1045642* was linked to a significant pain reduction under a combination therapy with morphine and nortriptyline (Table 3) [98].

**Safety.** The above-mentioned study of Benavides et al. also investigated the relationship between *ABCB1* rs1045642 and several side effects including sleepiness, constipation, and blood pressure. No noticeable linkage was identified (Table 3) [98]. CPIC recommends dose adjustments for nortriptyline according to the *CYP2D6* and *CYP2C19* phenotypes, in other words, *CYP2D6* UMs (increase of the target therapeutic dose), *CYP2D6* IMs (25% reduction in the recommended starting dose), *CYP2D6* PMs (50% reduction in the recommended starting dose if nortriptyline is warranted), and *CYP2C19* PMs (50% reduction in the recommended starting dose if nortriptyline is warranted) [89].

##### Imipramine

Imipramine is used for treating depression and also off-label for pain treatment. Imipramine is metabolized by CYP1A2, CYP3A4, CYP2C19, and CYP2D6 [99]. Imipramine is initially metabolized to desipramine by CYP2C19 with desipramine, and subsequently metabolized to a less active 2-hydroxyimipramine by CYP2D6 [100]. Individuals, who are *CYP2D6* PMs or *CYP2C19* PMs showed higher plasma concentrations compared to the *CYP2D6* or *CYP2C19* NMs [101,102]. Due to concerns about the possible adverse effects, the CPIC recommends a 50% reduction in the starting dose in PM patients with *CYP2D6* or *CYP2C19* [89]. Morinobu et al. reported that the *N*-demethylation of imipramine is impaired, as shown in five depressed PM patients with genetic defects in the *CYP2C19* gene, hinting to the importance of these SNPs for the safety of imipramine [103].

**Efficacy.** We identified one study that investigated the association between genetic polymorphisms in *CYP2D6* and the imipramine response in chronic low back pain treatment. Siegenthaler et al. observed that imipramine showed no significant effect on low back pain reduction in 50 patients in general [104]. In addition, Schliessbach et al. reported no clear differences between the *CYP2D6* genotypes and pain reduction [105]. Thus, the effect of imipramine in lower back pain is overall low. No associations of imipramine efficacy with the *CYP2D6* genotype could be detected (Table 3).

**Safety.** Imipramine is associated with multiple adverse effects including dry mouth, weight gain, drowsiness, and less frequent cardiac-related side effects [106]. No study was detected to have investigated the association between pharmacogenetic markers and the safety outcome with imipramine in vivo (Table 3). The DPWG suggests decreasing the imipramine dose in *CYP2C19* PMs due to the increased risk of side effects, whereas *CYP2C19* UMs and IMs can receive the full dose of the drug [96].

#### 4.2.2. Selective Serotonin Reuptake Inhibitors

SSRIs are drugs prescribed for major depressive and anxiety disorders and may also be used to treat psychiatric conditions such as obsessive-compulsive disorder. Of note, multiple studies found no significant effect of SSRIs in painful conditions such as fibromyalgia, migraine, and noncardiac chest pain, while their efficacy in neuropathic pain treatment has been scarcely studied [107,108,109,110]. SSRIs increase serotonergic activity by decreasing the presynaptic serotonin reuptake and are mainly metabolized by CYP2D6 and CYP2C19 [111,112].

##### Citalopram and Escitalopram 

Escitalopram is the *S*-enantiomer of the SSRI citalopram, which is a racemic mixture of enantiomers. Escitalopram is mainly used in the treatment of major depressive disorder. Additionally, data exist concerning some effects of escitalopram on neuropathic pain, however, it is not approved by the FDA or EMA for the indication of pain [113]. Citalopram and escitalopram are mainly metabolized by CYP2C19 and, to a lesser extent, by CYP3A4 and CYP2D6 [114]. Several studies have shown elevated plasma concentrations of citalopram and escitalopram in subjects who are *CYP2C19* PMs, which may increase the occurrence of adverse events [115,116,117]. Moreover, *CYP2C19* IMs may have higher citalopram and escitalopram plasma concentrations [118].

**Efficacy.** The association between various gene polymorphisms and escitalopram response in peripheral neuropathic pain treatment was assessed by Brasch-Andersen et al. [119]. Genetic variants in *HTR2A*, *HTR2C*, *CYP2C19*, and *ABCB1* were tested in 34 patients. A statistically significant difference was noticed only for the *HTR2C* variant rs6318 (i.e., a decreased pain relief in C allele carriers compared to G allele carriers). It has, however, to be noted that the sample size was limited and that this finding should be further investigated in higher powered studies (Table 3).

**Table 3 pharmaceutics-14-01190-t003:** Studies investigating the association between pharmacogenetics and the effect and safety of antidepressants in pain in vivo (human studies).

Study	Drug Class	Drug Name	Ethnicity	Study Design	FDA/EMA Status (Indication for Pain)	Outcome Measured	Gene Assessed	Variants	Findings
Wilder-Smith et al. (2005) [93]	TCA	Amitriptyline	N/A	30 patients with postamputation pain	Approved	Amitriptyline efficacy	*CYP2D6*	PMs and UMs	Lower pain levels in *CYP2D6* PMs
Chaudhry et al. (2017) [94]	TCA	Amitriptyline	Black African (*n* = 21), Caucasian (*n* = 9), Indian (*n* = 1)	31 patients with diabetic peripheral neuropathy	Approved	Amitriptyline treatment response and adverse effects	*CYP2D6*	**1/*1*, **1/*1xN*, **1/*45*, **2/*2*, **2M/*35*, **1/*2*, **35/*41*, **1/*17*, **2/*17*, **2/*4*, **2xN/*5*, **2/*29*, **1/*29*, **17/*84*	No effect of *CYP2D6* phenotype on amitriptyline efficacy. A trend towards more severe adverse effects in *CYP2D6* IMs compared to NMs
Benavides et al. (2021) [98]	TCA	Nortriptyline	Caucasian	25 neuropathic pain patients	Not approved	Nortriptyline treatment response and adverse effects	*CYP2C19* *CYP2D6* *ABCB1*	*CYP2C19 *2*, **3*, **4*, **5*, **6*, **7*, **8*, **17* *CYP2D6 rs1065852*, **2A*, **3*, **4*, **6*, **7*, **8*, **9*, **10*, **11*, **12*, **14*, **15*, *rs28371706*, **17*, **20*, **29*, **35*, **41*, *rs1135840*, **40*, **58*, **64* *ABCB1 rs1045642*, *rs2032582*	*ABCB1 rs1045642 C* homozygotes showed an improved therapy response in pain conditions under a combined therapy with nortriptyline and morphine
Siegenthaler et al. (2015) [104]	TCA	Imipramine	N/A	50 patients with chronic low-back pain	Not approved	Imipramine efficacy	*CYP2D6*	**6*, **7*, **8*, **10*, **41*, **3A*, **4*, **5*, **2*	No significant effect of amitriptyline on low back pain reduction
Schliessbach et al. (2018) [105]	TCA	Imipramine	N/A	50 patients with chronic low-back pain	Not approved	Imipramine response	*CYP2D6*	**1*, **3*, **4*, **5*, **6*, **8*, **10*, **41*	No overall reduction in low back pain with imipramine. No effect of *CYP2D6* phenotype on pain tests results
Brasch-Andersen et al. (2011) [119]	SSRI	Escitalopram	N/A	34 patients with peripheral neuropathic pain	Not approved	Escitalopram treatment response	*HTR2A*, *HTR2C*, *ABCB1*, *CYP2C19*, *SLC6A4*	*HTR2A rs6314 GG*, *GA*, *AA* *HTR2C rs6318 GG*, *GC*, *CC* (women) *HTR2C rs6318 G*, *C* (men) *ABCB1 rs2032582 GG*, *GT/AT*, *TT* *SLC6A4 5-HTTL* polymorphic region *L/L*, *L/S*, *S/S*	Little evidence for decreased pain relief in *HTR2C C* allele carriers in male participants
Aldrich et al. (2019) [120]	SSRI	Escitalopram	White Black Other	248 patients with depression and anxiety	Not approved	Escitalopram adverse effects	*CYP2C19*	*CYP2C19*1*, **2*, **3*, **4*, **5*, **6*, **7*, **8*, **17*	*CYP2C19* PMs and IMs showed a higher total number of side effects compared with NMs and UMs
Kuo et al. (2013) [121]	SSRI	Escitalopram	Chinese	158 patients with massive depressive disorder	Not approved	Escitalopram adverse effects	*CYP1A2*	*CYP1A2 rs2069521*, **1K*, **1F*, *rs4646425*, *rs35796837*, *rs34058039*, *rs2472304*, *rs3743484*, *rs4646427*, *rs2470890*	*CYP1A2* SNPs *rs2069521*, *rs4646425*, and *rs4646427* are associated with dry mouth, nausea, and vomiting at week 2, and fatigue at week 1

PM = poor metabolizer, UM = ultra-metabolizer, IM = intermediate metabolizer, NM = normal metabolizer, FDA = Food and Drug Administration, EMA = European Medical Agency, TCA = tricyclic antidepressant, N/A = not available, SSRI = selective serotonin reuptake inhibitor.

**Safety.** Aldrich et al. reported that *CYP2C19* PMs had the most and that UMs had the fewest side effects among the 254 pediatric patients with anxiety and depressive disorders [120]. Major adverse events caused by escitalopram included dry mouth, fatigue, nausea, and vomiting. Kuo et al. [121] detected that *CYP1A2* SNPs *rs2069521*, *rs4646425*, and *rs4646427* had a notable relation with dry mouth, fatigue, nausea, and vomiting. Moreover, this study showed that *CYP1A2* RMs probably experienced more severe side effects compared to the PMs (Table 3) [121]. A case report study is mentioned here, which reported a possible relation of serotonin syndrome and *CYP2C19* PM status. Schult et al. revealed that impaired *CYP2C19* activity may increase the plasma concentration and toxicity of citalopram [122]. A cross-sectional study by Castro et al., which included 9777 subjects who were treated with citalopram, revealed modest QT (QT time, ECG) prolongations with citalopram [123]. The FDA recommends a 50% citalopram dose reduction in patients who are *CYP2C19* PMs due to the risk of QT prolongation with the drug. The DPWG recommends avoiding escitalopram in *CYP2C19* UMs due to the decreased efficacy of this antidepressant (paroxetine or fluvoxamine are recommended for these patients). *CYP2C19* PMs have an increased risk of QT prolongation and life-threatening arrhythmias such as torsades de pointes tachycardia, which necessitates the prescription of escitalopram at 50% of the standard maximum dose. For *CYP2C19* IMs, the maximum dose should be 75% of the standard dose [124]. In contrast, *CYP2C19* IMs and *CYP2C19* extensive metabolizers (EMs) do not require dose adjustments according to the CPIC recommendations [112].

## 5. Discussion

We systematically explored and summarized the relevant human in vivo studies that investigated genetic polymorphisms considering the safety and efficacy of important drugs used in pain management. We laid our focus specifically on anti-inflammatory drugs and antidepressants, which are widely used in clinical practice for pain therapy in the framework of many diseases. Our review identified relevant medication–gene interactions for nine drugs involving especially different NSAIDs and genes including *CYP2C8*, *CYP29*, *UGT2B7*, *ABCC2*, and *CHST2* and, to a lesser extent, ADs involving *CYP2C19*, *CYP2D6*, *CYP1A2*, and *HTR2C*. 

For drugs belonging to the class of NSAIDs, it was noted that the pharmacogenetic studies especially focused on the role of polymorphically expressed genes belonging to the *CYP* family (i.e., *CYP2C9* and, to a lesser extent, of *CYP2C8*). Studies indicate that genetic variants leading to PM genotypes and phenotypes of *CYP2C9* (i.e., the most often studied SNP *CYP2C9*3* and the less often studied variant *CYP2C9*2*) have a clear influence on the in vivo pharmacokinetics of the majority of therapeutics discussed in the review (i.e., ibuprofen, celecoxib, diclofenac, and meloxicam). Interestingly, this finding is not directly associated with shifts in the efficacy of these drugs in pain treatment. We identified many studies that investigated the impact of the mentioned *CYP* variants on the pharmacokinetics of NSAIDs. However, the number of studies investigating the impact of these variants on pain relief are overall very limited, detecting only a very modest impact of the *CYP2C8* and *CYP2C9* genetic variants on the NSAID therapeutic effect. Human in vivo studies investigating the association between the safety of NSAIDs and a polymorphic expression of drug metabolizing enzymes can more often be found. The studies identified here for several drugs such as ibuprofen, diclofenac, or meloxicam showed a relationship between the PM genotypes and an increased risk for the important side effects such as GI bleeding or cardiovascular events. In the case of aspirin, variants in *UGT1A6* (*UGT1A6*2/*2*) have been repeatedly associated with a changed pharmacokinetics of the drug, which is, however, not easily translated into consequences for the effect or safety of the drug. GI bleeding remains one of the most important adverse effects of aspirin and NSAIDs. Among the patients taking aspirin or NSAIDs, allelic variants in the genes *TNF-α* and *VKORC1* have been associated with a higher risk of UGIB compared to carriers of other alleles. Despite these data, the number of in vivo studies focusing on aspirin safety or efficacy in relation to pharmacogenetics is still very limited, calling for aspirin-related pharmacogenetic in vivo studies in the future. 

Studies investigating TCAs indicate a better efficacy of amitriptyline in neuropathic pain reduction in *CYP2D6* PMs. Furthermore, one study detected a link between *ABCB1 rs1045642* and significant pain reduction under a combination therapy with nortriptyline and morphine. Genetic alterations in the *CYP2C19* gene and in *CYP2D6* may be predictive for the risk of adverse effects for imipramine and amitriptyline, respectively, as explored in several studies including depressive patients. Similarly, polymorphisms of *CYP2C19* and *CYP1A2* have been shown to be associated with an increased risk for the characteristic side effects of escitalopram. We detected only one study that indicated a potentially relevant role of *HTR2C* SNP *rs6318* in the efficacy of pain treatment with SSRIs, leading to the conclusion that the knowledge about the associations between genetic variants regarding changes in the efficacy of antidepressants in pain therapy is currently very limited. 

In conclusion, it is noted that well-powered human in vivo studies assessing the pharmacogenetic impact of relevant genes in pain patients treated with NSAIDs or antidepressants are lacking. This lack is especially abundant for efficacy estimations of these drugs in relation to genetic polymorphisms in pain management. The few studies found in the context of efficacy hint to a rather poor correlation between shifts in the pharmacokinetics and efficacy consequences with NSAIDs in the treatment of pain. *CYP2C19* and *CYP2D6* may have an impact on the efficacy of TCAs in pain management. Although overall rather limited in number, these studies indicate a higher risk for severe side effects such as GI bleeding or heart-related side effects in the case of PM genotypes of *CYP2C9* treated with NSAIDs. The studies included in our systematic review are based on cohorts of patients of different ethnicities, which may create certain difficulties and limitations in interpreting and comparing results. While studies regarding ibuprofen response [37] and aspirin adverse effects [83] only included White subjects, studies on amitriptyline treatment response [94] and escitalopram adverse effects [120] include different ethnicities such as Black Africans, White, and Indian subjects. For more accurate and complete analyses of pharmacogenetic influences on drug safety and efficacy, clinical studies involving larger groups of patients of the same ethnic group are needed in the future to be able to even more accurately describe biomarker dependent drug safety and efficacy in different ethnicities. A bias assessment considering aspects such as the randomization of the studies, the Hardy–Weinberg equilibrium, and the overall size of the studies did reveal some additional limitations regarding the conclusiveness of the pharmacogenetic impact in the field of pain management. While the vast majority of studies that we detected were randomized, only 10 out of the 25 papers discussed in our review mentioned or tested for the Hardy–Weinberg equilibrium. Eight of the articles were in equilibrium and two of them reported a slight deviation. Additionally, 13 studies included less than 100 participants, which further emphasizes the general problem of the small sample sizes included in the studies thus far. These aspects have to be optimized in future studies investigating the pharmacogenetic influences on the efficacy and safety of antidepressants and NSAIDs in pain management to obtain more conclusive results. Further in vivo studies are needed to consolidate the role of relevant polymorphisms in the safety of NSAIDs and to further elucidate the role of pharmacogenetics regarding the efficacy of NSAIDs and antidepressants in pain management.

## Figures and Tables

**Figure 1 pharmaceutics-14-01190-f001:**
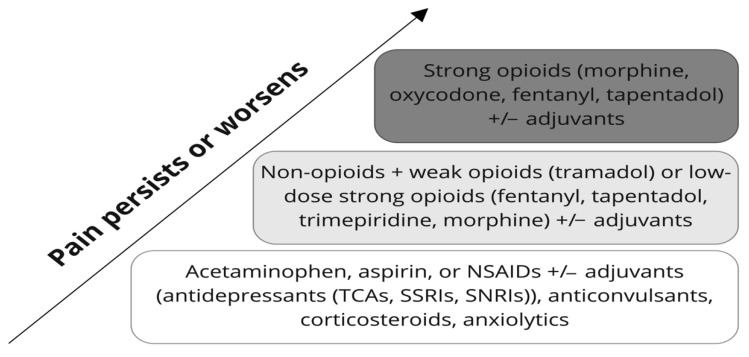
The WHO analgesic ladder. The “three step” concept of medication-assisted pain management was originally developed by the WHO in 1986 for cancer-associated pain treatment but is now widely used in all areas of medicine. The three key principles of pain treatment according to the WHO are “by the clock, by the mouth, by the ladder”, which means regular and timely (“by the clock”) use of the safest and simplest forms (“by the mouth”) of the most effective and safest drugs, starting with less active non-opioid analgesics (with or without different adjuvants such as antidepressants (TCAs, SSRIs, SNRIs), anticonvulsants, corticosteroids, anxiolytics) with a gradual transition to more active opioid drugs with more adverse effects (“by the ladder”).

**Figure 2 pharmaceutics-14-01190-f002:**
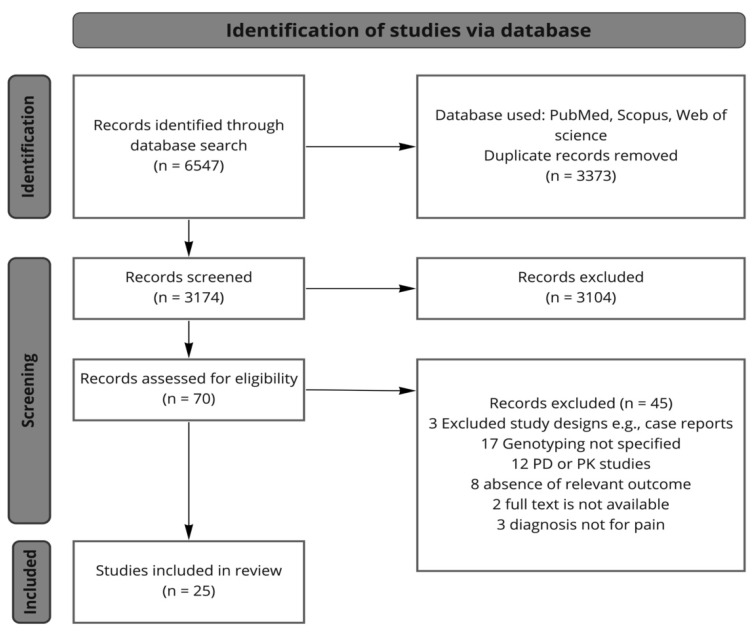
The study selection flow chart. PD = pharmacodynamic, PK = pharmacokinetic.

**Table 1 pharmaceutics-14-01190-t001:** The specific drugs included in the search.

Drug Class	Drug Name
Non-steroidal anti-inflammatory drugs (NSAIDS)	Aspirin, celecoxib, etoricoxib, parecoxib, diclofenac, aceclofenac, ibuprofen, dexibuprofen, indomethacin, acemetacin, ketoprofen, dexketoprofen, meloxicam, piroxicam, naproxen, oxaprozine, ketorolac, nabumetone, metamizole, phenazone, propyphenazone, tiaprofenic acid
Tricyclic antidepressants (TCAs)	Amitriptyline, amoxapine, clomipramine, desipramine, doxepin, imipramine, nortriptyline, protriptyline, trimipramine
Serotonin-norepinephrine reuptake inhibitors (SNRIs)	Duloxetine, venlafaxine, desvenlafaxine, levomilnacipran
Selective serotonin reuptake inhibitors (SSRIs)	Citalopram, escitalopram, fluoxetine, fluvoxamine, paroxetine, sertraline
Tetracyclic antidepressants	Maprotiline, mianserin, mirtazapine, setiptiline
Monoamine oxidase (MAO) inhibitors	Isocarboxazid, phenelzine, selegiline, tranylcypromine

## Data Availability

Not applicable.

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
