# Peer review of "Pharmacogenetics and Pain Treatment with a Focus on Non-Steroidal Anti-Inflammatory Drugs (NSAIDs) and Antidepressants: A Systematic Review"

_pharmaceutics, 2022, doi:10.3390/pharmaceutics14061190_

Round 1
Reviewer 1 Report
The authors conducted a systematic review summarizing the impact of pharmacogenetics on the effect and safety of non-steroidal anti-inflammatory drugs (NSAIDs) and antidepressants for pain treatment. The research question is valid, and the study design is adequate. The results are clearly and carefully presented by the authors. I have some specific minor questions.
Minor suggestions
As I commented above, the systematic review design seemed to be adequate. However, I missed any mention of publication bias assessment. Did the authors assess if any of the 25 eligible papers could have some pertinent bias that may affect the interpretation of their conclusions? For example, genetic association studies (which were a prerequisite for inclusion in the analysis) often have low power and perhaps most negative results are not even published. Please elaborate. If necessary, include some sentences in the Discussion section addressing this limitation.
On the same topic of genetic association studies: did the authors assess if the eligible studies correctly reported allele frequencies and checked whether any violation of Hardy-Weinberg occur? In other words, the genotyping results of each study had good quality overall?
Page 4, lines 166 and 167. The authors mention "A study had to (1) be written in English". However, the flowchart (Figure 2) states that two studies were excluded due to "unable to translate". Therefore, one could imply that some studies were translated from some other language into English and some were not. Please address this discrepancy.
Author Response
Comments and Suggestions for Authors
The authors conducted a systematic review summarizing the impact of pharmacogenetics on the effect and safety of non-steroidal anti-inflammatory drugs (NSAIDs) and antidepressants for pain treatment. The research question is valid, and the study design is adequate. The results are clearly and carefully presented by the authors. I have some specific minor questions.
Minor suggestions
As I commented above, the systematic review design seemed to be adequate. However, I missed any mention of publication bias assessment. Did the authors assess if any of the 25 eligible papers could have some pertinent bias that may affect the interpretation of their conclusions? For example, genetic association studies (which were a prerequisite for inclusion in the analysis) often have low power and perhaps most negative results are not even published. Please elaborate. If necessary, include some sentences in the Discussion section addressing this limitation.
Answer of the authors:
Thank you for the remark. The following section has been added to the discussion part:
“A bias assessment considering aspects such as randomization of the studies, Hardy Weinberg equilibrium and the overall size of the studies did reveal some additional limitations regarding the conclusiveness of the pharmacogenetic impact in the field of pain management. While the vast majority of studies that we detected were randomized, only 10 out of 25 papers discussed in our review, mentioned or tested for Hardy Weinberg equilibrium. Eight of the articles, were in equilibrium and 2 of them reported a slight deviation. Additionally, 13 studies included less than 100 participants which further emphasises the general problem of small sample sizes included in the studies so far. These aspects have to optimized in future studies investigating pharmacogenetic influences on the efficacy and safety of antidepressants and NSAIDs in pain management to obtain more conclusive results.”
On the same topic of genetic association studies: did the authors assess if the eligible studies correctly reported allele frequencies and checked whether any violation of Hardy-Weinberg occur? In other words, the genotyping results of each study had good quality overall?
Answer of the authors:
Thank you for the nice remark. Ten out of 25 papers mentioned in our review discussed Hardy Weinberg equilibrium. Eight of the articles, were in equilibrium and 2 of them reported a slight deviation. We have added now additional information in the discussion mentioning these aspects as limiting factors that have to become better in future studies along with further information on the size of the studies (see also the answer to your former question).
Page 4, lines 166 and 167. The authors mention "A study had to (1) be written in English". However, the flowchart (Figure 2) states that two studies were excluded due to "unable to translate". Therefore, one could imply that some studies were translated from some other language into English and some were not. Please address this discrepancy.
Answer of the authors:
Thank you for the remark. This was a mistake. These studies were excluded due to the absence of full text. We have corrected this mistake.
Reviewer 2 Report
In the paper authors report a systematic review on the pharmacogenetics of drugs used in pain treatment. The literature search has been well conducted and results are clearly reported.
Some further detail may improve the paper and the presentation of the study:
In the paper authors cite international guidelines of Clinical Pharmacogenetics Implementation Consortium (CPIC), but no citation about DPWG (Dutch Pharmacogenetics Working Group - PharmGKB) guidelines are reported. It can be interesting to cite both, CPIC and DPWG guidelines, especially in cases where the suggestions are different (i.e. escitalopram)
Considering ethnic differences, the discussion section must be improved citing the hard comparison between studies conducted on different populations.
Author Response
In the paper authors report a systematic review on the pharmacogenetics of drugs used in pain treatment. The literature search has been well conducted and results are clearly reported.
Some further detail may improve the paper and the presentation of the study:
- In the paper authors cite international guidelines of Clinical Pharmacogenetics Implementation Consortium (CPIC), but no citation about DPWG (Dutch Pharmacogenetics Working Group - PharmGKB) guidelines are reported. It can be interesting to cite both, CPIC and DPWG guidelines, especially in cases where the suggestions are different (i.e. escitalopram).
Answer of the authors:
Many thanks for the suggestions. Based on the reviewer suggestions above We have added the following sections into the manus:
Imipramine
…The DPWG suggests adjusting the imipramine dose in CYP2C19 PM due to the increased risk of side effects, whereas CYP2C19 UM and IM can receive the full dose of the drug.
Escitalopram
…The DPWG recommends avoiding escitalopram in CYP2C19 UM due to a decreased efficacy of this antidepressant (paroxetine or fluvoxamine are recommended for these patients). CYP2C19 PM have an increased risk of QT prolongation and life-threatening arrhythmias such as torsades de pointes tachycardia, which leads to the recommendation to prescribe escitalopram at 50% of the standard maximum dose. For CYP2C19 IM, the maximum dose should be 75% of the standard dose. In contrast to this, CYP2C19 IM and CYP2C19 EM do not require dose adjustments according to CPIC recommendations.
Amitryptiline
…The DPWG recommends using increased doses of amitriptyline in CYP2D6 UM because of the higher metabolic rate of this drug, whereas CYP2D6 IM and PM should receive 75% and 70% of the standard dose.
- Considering ethnic differences, the discussion section must be improved citing the hard comparison between studies conducted on different populations.
Answer of the authors:
Thank you for the comment. We have added following section into the discussion part:
The studies included in our systematic review are based on cohorts of patients of different ethnicities, which may create certain difficulties and limitations in interpreting and comparing results. While e.g., studies regarding ibuprofen response [37] and aspirin adverse effects [83] only included white subjects, studies on amitriptyline treatment response [94], and escitalopram adverse effects [120], include different ethnicities such as black Africans, white and Indian subjects. For more accurate and complete analyses of pharmacogenetic influences on drug safety and efficacy, clinical studies involving larger groups of patients of the same ethnic group are needed in the future to be able to even more accurately describe biomarker dependent drug safety and efficacy in different ethnicities.
Reviewer 3 Report
Although there seem to be no major issues, there are way too many syntax issues that need to be taken care of so that the readers can fully comprehend the article (see below).
1) Please replace "Sweden." with "Sweden" (line 6).
2) Please provide city name for "Advanced Molecular Technology" (line 7).
3) Please define abbreviation for "LLC" (line 7), "PRISMA" (line 12), "EMA" (line 456), "FDA" (line 457), "QT" (line 584).
4) Please use consistent font formatting for ":" in "Abstract:" (line 9).
5) Please format "th" in "18th" using superscript (lines 14, 159).
6) Please change "6547" to "6,547" (lines 15, 186).
7) Please replace "ibuprofen/ CYP2C9, (2) celecoxib/ CYP2C9; (3) piroxicam/ CYP2C8, CYP2C9; (4) diclofenac/CYP2C9, UGT2B7, CYP2C8, ABCC2; (5) meloxicam/ CYP2C9; (6) aspirin/ CYP2C9, SLCO1B1 and CHST2; 7) amitriptyline/ CYP2D6 and CYP2C19; (8) imipramine/ CYP2C19; (9) nortriptyline CYP2C19, CYP2D6, ABCB1 and (10) escitalopram and HTR2C, CYP2C19 and" with "ibuprofen/CYP2C9, (2) celecoxib/CYP2C9; (3) piroxicam/CYP2C8, CYP2C9; (4) diclofenac/CYP2C9, UGT2B7, CYP2C8, ABCC2; (5) meloxicam/CYP2C9; (6) aspirin/CYP2C9, SLCO1B1, and CHST2; 7) amitriptyline/CYP2D6 and CYP2C19; (8) imipramine/CYP2C19; (9) nortriptyline/CYP2C19, CYP2D6, ABCB1; and (10) escitalopram/HTR2C, CYP2C19, and"
8) Please replace "7)" with "(7)" (line 19).
9) Please format "ABCB1" consistently with the rest of the text (line 20).
10) Please remove italics formatting from "in vivo" (lines 21, 23, 286, 314, 333, 548, 591, 604, 611, 621, 623, 636, 645).
11) Please change "globally,[1]" to "globally [1]" (line 30).
12) Please change "(IASP)" to "(IASP) (www.iasp-pain.org)" (line 32).
13) Please remove hyperlink from "IASP" (line 32).
14) Please replace "damage.”" with "damage”." (line 34).
15) Please replace "perspective" with "perspective," (line 35).
16) Please change "types" to "types," (line 35).
17) Please replace "pathological pain" with "pathological" (line 35).
18) Please change "type. [2]" to "type [2]." (line 37).
19) Please replace "small, and" with "small and" (line 38).
20) Please replace "pain. [1]" with "pain [1]." (line 38).
21) Please change "pain, such" to "pain such" (line 39).
22) Please replace "e.g.," with "e.g." (lines 39, 103).
23) Please change "men. [3,4]" to "men [3,4]." (line 41).
24) Please change "18-25" to "18–25" (lines 42, 44).
25) Please replace "have" with "has" (line 42).
26) Please change "66.9 %" to "66.9%" (line 42).
27) Please replace "group. [5,6]" with "group [5,6]." (line 44).
28) Please change "pain and the mechanisms are many" to "pain" (line 45).
29) Please change "treatment. [7]" to "treatment [7]." (line 49).
30) Please replace "dependence. [8]" with "dependence [8]." (line 52).
31) Please change "Furthermore" to "Furthermore," (line 52).
32) Please replace "3.5-year" with "3.5 year" (line 53).
33) Please change "infarction. [9]" to "infarction [9]." (line 54).
34) Please replace "gastrointestinal" with "gastrointestinal (GI)" (line 54), "gastrointestinal" with "GI" (lines 248, 288, 336, 412, 615, 644), "Gastrointestinal" with "GI" (line 618), and "gastrointestinal (GI)" to "GI" (line 379).
35) Please replace "bleeding. [10]" with "bleeding [10]." (line 55).
36) Please change "somnolence" to "somnolence," (line 55).
37) Please change "antidepressants. [11]" to "antidepressants [11]." (line 56).
38) Please replace "drugs. [12]" with "drugs [12]." (line 58).
39) Please change "patients. [13]" to "patients [13]." (line 60).
40) Please replace "receptors" with "receptors," (line 61).
41) Please remove italics formatting from "CYP450" (line 67).
42) Please remove italics formatting from "CYP2D6" (line 68).
43) Please change "codeine that" to "codeine, which" (line 68).
44) Please change "results" to "result" (line 70).
45) Please replace "while" with "while," (line 71).
46) Please replace "effect. [14,15]" with "effect [14,15]." (line 72).
47) Please change "opioids. [16]" to "opioids [16]." (line 73).
48) "Although NSAIDs and antidepressants are important pillars in current treatment strategies of pain, associations between pharmacogenetically relevant SNPs and changes in safety and efficacy of these drug groups is even much more unclear" (line 73) does not seem to make sense with respect to "is even much more unclear". More unclear than what?
49) Please replace "which" with "that" (line 77).
50) Please change "anti-inflammatory" to ""anti-inflammatory drugs" (line 80).
51) Please replace "medicine. [17]" with "medicine [17]." (line 86).
52) Please change "[18]" to "[18]." (line 88).
53) Please replace "and suggests" with "while it suggests" (line 90).
54) Please replace "necessary. [19,20]" with "necessary [19,20]." (line 91).
55) Please change "relief" to "relief," (line 92).
56) Please replace "financial" with "financial," (line 93).
57) Please change "capabilities. [18]" to "capabilities [18]." (line 93).
58) Please replace "methods, but" with "methods but" (line 95).
59) Please change "example, surgical" to "example surgical" (line 95).
60) Please change "[20]" to "[20]." (line 96).
61) Please replace "work" with "work," (line 96).
62) Please change "in the first step of pharmacological pain treatment acetaminophen should be used" to "acetaminophen should be used in the first step of pharmacological pain treatment" (line 99).
63) Please replace "avoiding" with "while avoiding" (line 100).
64) Please change "(in pain conditions with a relevant of inflammatory component, or in case of metastatic bone lesions). [19–22]" to "(in pain conditions with a relevant inflammatory component or in case of metastatic bone lesions) [19–22]." (line 101).
65) Please replace "diclofenac, indomethacin" with "diclofenac or indomethacin" (line 103).
66) Please replace "patient. [23]" with "patient [23]." (line 106).
67) Please change "(neuropathic pain, fibromyalgia, neuralgia)" to "(neuropathic pain, fibromyalgia, or neuralgia)," (line 106).
68) Please replace "Tricyclic" with "tricyclic" (line 108).
69) Please change "serotonin-norepinephrine" to "and serotonin-norepinephrine" (line 109).
70) Please change "and/or" to either "and" or "or" (lines 110, 116, 161, 575).
71) Please replace "anticonvulsants. [24,25]" with "anticonvulsants [24,25]." (line 110).
72) Please change "patient: in" to "patient. In" (line 112).
73) Please replace "(denosumab, tanezumab)" with "(denosumab, tanezumab)," (line 114).
74) Please change "used. [26]" to "used [26]." (line 115).
75) Please replace "infarction, severe" with "infarction or severe" (line 116).
76) Please replace "latter mentioned cases. [27]" with "the latter mentioned cases [27]." (line 118).
77) Please change "(fentanyl, tapentadol, trimepiridine, morphine). [19–22]" to "(fentanyl, tapentadol, trimepiridine, or morphine) [19–22]." (line 121).
78) It is not clear what the authors mean by "the minimum effective opioid dose will last if possible" in "Opioids should be added to treatment gradually and may be used in combination with first-stage medications (NSAIDs and paracetamol can potentiate the effects of opioids) with the expectation that the use of the minimum effective opioid dose will last if possible" (line 122)?
79) Please replace "last if possible. [19,20,28]" with "last [19,20,28]." (line 125).
80) Please change "tapentadol" to "or tapentadol" (line 127).
81) Please change "used. [19–22]" to "used [19–22]." (line 127).
82) Please replace "opioids, and" with "opioids and" (line 128).
83) Please change "effects. [19,20,29]" to "effects [19,20,29]." (line 129).
84) Please replace "Adjuvants" with "adjuvants" 2x, "OR" with "or", "+/OR" with "or", "SSRI, SNRI" with "SSRIs, SNRIs" in Figure 1.
85) Please remove italics formatting from "Figure 1." (line 131).
86) Please change "Analgesic Ladder" to "analgesic ladder" (line 131).
87) Please replace "steps" with "step" (line 131).
88) Please change "adjuvants, such as" to "adjuvants such as" (line 136).
89) Please replace "SSRI, SNRI" with "SSRIs, SNRIs" (line 137).
90) "To catch a wide as possible range of papers the initial protocol was widened adding the additionally search terms “Genetic polymorphism” OR “Safety” OR “Efficacy “AND/OR the drug names listed in [Table 1]" (line 156) does not seem to be grammatically correct with respect to "catch a wide as possible range" (line 156). Please fix.
91) Please change "papers" to "papers," (line 156).
92) Please replace "additionally" with "additional" (line 157).
93) Please change "[Table 1]" to "Table 1" (line 158).
94) Please replace "18th, 2021" with "18th 2021" (line 159).
95) Please change "abstracts" to "abstracts," (line 160).
96) Please replace "(3) and be" with "(3), and represent" (line 167).
97) Please replace "step" with "step," (line 168).
98) Please change "(2) human" to "and (2) represent human" (line 176).
99) Please replace "review" with "review," (line 180).
100) Please change "screening: Same" to "screening, same" (line 183).
101) Please change "Scopus" to "Scopus," (line 186).
102) Please replace "criteria that" with "criteria" (line 187).
103) Please change "review. (Figure 2)" to "review (Figure 2)" (line 188).
104) Please change "n= 6547" to "n = 6,547", "n=3174" to "n = 3,174", "n= 70" to "n = 70", "n=25" to "n = 25", "n=3373" to "n = 3,373", "n= 3104" to "n = 3,104", "n= 45" to "n = 45" in Figure 2.
105) From the legend to Figure 2 is not clear what does "PD" and "PK" stand for?
106) Please remove italics formatting from "Figure 2." (line 194).
107) Please replace "i.e., NSAIDs" with "i.e. NSAIDs" (line 197).
108) Please replace "SSRI" with "SSRIs" (line 197).
109) Please change "drugs (NSAIDs)" to "drugs" (line 200).
110) Please replace "NSAID" with "NSAIDs" (line 202).
111) Please change "antipyretic" to "antipyretic," (lines 203, 390).
112) Please replace "Cytochrome P450 enzyme family, such" with "cytochrome P450 enzyme family such" (line 205).
113) Please change "CYP2C8" to "CYP2C8," (line 208).
114) Please change "UGT1A4 and UGT1A9, also" to "UGT1A4, and UGT1A9 also" (line 207).
115) Please replace "NSAIDs’" with "NSAID" (line 207).
116) Please change "single nucleotide polymorphisms (SNPs)" to "SNPs" (line 208).
117) Please change "response. (Table 2)" to "responses (Table 2)." (line 209).
118) Please replace "COX-2" with "COX-2," (line 212).
119) Please change "formation. [33]" to "formation [33]." (line 213).
120) Please replace "S- (+) and R-(-)" with "S-(+) and R-(−)" (line 213).
121) Please format "S" and "R" in "S- (+) and R-(-)" using italics (line 213).
122) Please format "S" in "S-ibuprofen" using italics (lines 214, 215, 219 2x, 222, 224).
123) Please format "R" in "R-ibuprofen" using italics (lines 214, 220, 222, 223).
124) Please format "S" and "R" in "S- and R-enantiomers" using italics (line 215).
125) Please change "extent" to "extent," (lines 216, 458).
126) Please format "R" in "R-enantiomer" using italics (line 216).
127) Please replace "pharmacokinetics, as" with "pharmacokinetics as" (line 223).
128) Please change "efficacy as summarized in the following" to "efficacy" (line 227).
129) Please replace "[36] ," with "[36]," (line 229).
130) Please change "poor metabolizers" to "PMs" (lines 230, 483, 625).
131) Please change "intermediate/normal metabolizers (IM/NM)" to either "intermediate (IM) and normal (NM) metabolizers" or "intermediate (IM) or normal (NM) metabolizers" (line 231).
132) Please replace "However" with "However," (line 231).
133) Please change "on" to "in" (line 234).
134) Please replace "According to the data above" with something like "In summary" (line 239).
135) Please replace "efficacy. (Table 2)" with "efficacy (Table 2)." (line 240).
136) Please change "CYP2C9, CYP2C8" to "CYP2C9 and CYP2C8" (line 242).
137) Please replace "as also" with "as" (line 243).
138) Please replace "study" with "study," (line 244).
139) Please replace "seven" with "7" (line 245).
140) Please change "detected. [35]" to "detected [35]." (line 246).
141) Please replace "al" with "al." (lines 247, 281, 352, 360, 460, 468, 485, 504, 541, 582).
142) Please replace "indomethacin. [39] (Table 2)" with "indomethacin (Table 2) [39]." (line 249).
143) Please change "CPIC" to "Clinical Pharmacogenetics Implementation Consortium (CPIC)" (line 250) and "Clinical Pharmacogenetics Implementation Consortium (CPIC)" to "CPIC" (line 533).
144) "activity score" is not precisely defined in "The CPIC defines an activity score ranging from 0 to 2 based on the CYP2C9 diplotype status and recommends to start ibuprofen with the lowest recommended dose accompanied with monitoring of adverse effects in case of IMs (activity score 1) or with 25-50% of the lowest recommended dose in case of a CYP2C9 PM status to avoid side effects. [40]" (line 250). Does 2 relate to the most and 0 to the least severe disease phenotype or vice versa? Moreover, what does score of 1 indicate?
145) "The CPIC defines an activity score ranging from 0 to 2 based on the CYP2C9 diplotype status and recommends to start ibuprofen with the lowest recommended dose accompanied with monitoring of adverse effects in case of IMs (activity score 1) or with 25-50% of the lowest recommended dose in case of a CYP2C9 PM status to avoid side effects. [40]" (line 250) is too long. Please split into two sentences.
146) Please change "25-50%" to "25–50%" (line 253).
147) Please replace "effects. [40]" with "effects [40]." (line 254).
148) Please change "to" to "to a" (line 258).
149) Please replace "[43]" with "[43]," (line 259).
150) Please replace "UDP- glucuronosyltransferases (UGTs). [44]" with "UGTs [44]." (line 260).
151) Please change "[46] ," to "[46]," (line 265).
152) Please replace "i.e., pain" with "i.e. pain" (line 269).
153) Please replace "date as summarized in the following" with "date" (line 269).
154) Please change "detected" to something like "identified" (lines 271, 538, 612).
155) Please replace "participants, that" with "participants that" (line 277).
156) Please change "wild type" to something like "wild-type form" (line 278).
157) Please replace "efficacy, and" with "efficacy and" (line 280).
158) Please replace "patients with CYP2C9*1 /CYP2C9*3 genotype (IM)" with "IM patients with CYP2C9*1/CYP2C9*3 genotype" (line 282).
159) Please change "PM. [50] (Table 2)" to "PM patients (Table 2) [50]." (line 283).
160) Please replace "between" with "between the" (line 284).
161) Please change "report, Gupta" to "report Gupta" (line 287).
162) Please change "24-year-old" to "24 year old" (line 288).
163) Please replace "usage. [51] (Table 2)" with "usage (Table 2) [51]." (line 290).
164) It is not exactly clear what the authors mean by "dose titrate"..."individuals" in "The CPIC recommends to initiate the therapy with the lowest recommended dose and dose titrate and monitor individuals for side effects who are CYP2C9 IMs" (line 290)?
165) Please change "PMs a 50-75%" to "PMs, a 50–75%" (line 292).
166) Please replace "recommended. [40]" with "recommended [40]." (line 293).
167) Please change "class" to "class," (line 295).
168) Please remove italics formatting from "CYP2C9" (line 297).
169) Please remove italics formatting from "CYP2C8" (line 298).
170) Please replace "and to a lesser extent" with "and, to a lesser extent," (lines 297, 597, 601).
171) Please change "carriers. [52]" to "carriers [52]." (line 300).
172) Please replace "trials. (Table 2)" with "trials (Table 2)." (line 306).
173) "[53] Calvo et al., who studied the relation between CYP2C8 and CYP2C9 genotypes and the clinical efficacy of oral piroxicam as mentioned above, detected two subjects among 102 individuals with adverse reactions, with one carrying the CYP2C8*3 mutant and the CYP2C9*1/*3 genotype" (line 308) does not seem to be entirely correct as "Calvo et al." is not mentioned in the preceding text (above).
174) Please change "[53] Calvo et al." to "Calvo et al. [53]" (line 308).
175) Please replace "reactions, with" to either "reactions," or "reactions with" (line 311).
176) Please change "(carrying CYP2C8*3 mutant and the CYP2C9*1/*3 genotype). [53]" to "(carrying CYP2C8*3 mutant and the CYP2C9*1/*3 genotype) [53]." (line 313).
177) Please replace "collected" with "collected," (line 314).
178) Please change "manner. (Table 2)" to "manner (Table 2)." (line 315).
179) Please replace "PMs. [40]" with "PMs [40]." (line 317).
180) Please format "S" in "S-(+) enantiomer" using italics (line 320).
181) Please change "R-(-)" to "R-(−)" (line 321).
182) Please format "R" in "R-(-) enantiomer" using italics (line 321).
183) Please replace "enzymes, including" with "enzymes including" (line 325).
184) Please split "The obtained data may be explained by little involvement of CYP isozymes in ketoprofen metabolism, moreover the assessed UGT1A1 is not the main enzyme in ketoprofen glucuronidation [58], which may also be the reason of observed results" (line 329) into two sentences.
185) Please change "of" to "for the" (line 331).
186) Please replace "drug. (Table 2)" with "drug (Table 2)." (line 335).
187) Please change "vomiting" to "vomiting," (line 337).
188) Please change "frequent. [54]" to "frequent [54]." (line 338).
189) Please replace "Mejía-Abril" with "Mejía-Abril et al." (line 339).
190) Please change "trials. ] [57](Table 2)" to "trials (Table 2) [57]." (line 343).
191) Please replace "twenty-four patients. [64]" with "24 patients [64]." (line 353).
192) Please remove superscript formatting from "47" in "[47]" (line 355).
193) Please replace "outcome. (Table 2)" with "outcome (Table 2)." (line 358).
194) Please change "of" to "of the" (line 359).
195) Please replace "twenty-four patient" with "24 patients" (line 361).
196) Please change "metabolites. [67] (Table 2)" to "metabolites (Table 2) [67]." (line 364).
197) Please replace "COX 2" with "COX-2" (line 368).
198) Please change "provide" to something like "hint at" (line 376).
199) Please replace "conditions treatment. (Table 2)" with "treatment (Table 2)." or "management (Table 2)." (line 378).
200) Please change "Meloxicam" to "meloxicam" (line 379).
201) Please replace "ulceration" with "or ulceration" (line 380).
202) Please change "consisted of 22 healthy volunteers and was" to "was" (line 382).
203) Please change "above. [69] (Table 2)" to "above (Table 2) [69]." (line 384).
204) Please replace "meloxicam and" with something like "meloxicam treatment of" (line 385).
205) Please change "state" to "state is" (line 386).
206) Please replace "considered. [40]" with "considered [40]." (line 388).
207) Please replace "determined. (Table 2)" with "determined (Table 2)." (line 403).
208) Please change "bleeding. [77]" to "bleeding [77]." (line 404).
209) Please replace "et al., detected" with "et al. detected" (line 405).
210) Please change "groups; patients" to "groups, patients" (line 407).
211) Please replace "n=111" with "n = 111" (line 408).
212) Please change "(n=45)" to "(n = 45)," (line 408).
213) Please replace "n=482" with "n = 482" (line 408).
214) Please change "ulcer. [78]" to "ulcer [78]." (line 410).
215) Please replace "variant. [79] (Table 2)" with "variant (Table 2) [79]." (line 413).
216) Please provide reference for "Wang et al., investigated the association between genetic variants of tumor necrosis factor α (TNF-α) and upper gastrointestinal bleeding (UGIB)" (line 414).
217) Please change "et al., investigated" to "et al. investigated" (line 414).
218) Please replace "TNF- α" with "TNF-α" (line 417).
219) Please change "[80]" to "[80]." (line 421).
220) Please replace "Groza et al., tested" with "Groza et al. tested" (line 422).
221) Please change "NSAIDs/aspirin-induced" to "NSAID- or aspirin-induced" (line 424).
222) Please replace "UGIB. [81]" with "UGIB [81]." (line 426).
223) Please provide reference for "Piazuelo et al., examined the association between UGIB and nitric oxide synthase (eNOS) (a and b allele) and the platelet glycoprotein (GPIIIa) (PIA1 and PIA2 allele) genes" (line 426).
224) Please change "et al., examined" to "et al. examined" (line 426).
225) Please replace "UGIB" with "UGIB," (line 429).
226) Please change "aspirin[82]" to "aspirin [82]" (line 432).
227) Please replace "n= 94" with "n = 94" (Martinez et al. (2004) [39]), "n=124" with "n = 124" (Martinez et al. (2004) [39]), "CYP2B6, CYP2C8, CYP2C9, CYP2C19,CYP2D6,CYP3A4, PTGS2" with "CYP2B6 CYP2C8 CYP2C9 CYP2C19 CYP2D6 CYP3A4 PTGS2" (Saiz-Rodriguez et al. (2021) [36]), "NMs and UMs" to "NMs, and UMs" 2x (Saiz-Rodriguez et al. (2021) [36]), "IMs/NMs." with "IMs/NMs" (Saiz-Rodriguez et al. (2021) [36]), "CYP2C8, CYP2C9" with "CYP2C8 CYP2C9" (Weckwerth et al. (2020) [37]) in Table 2.
228) Please add "N/A" to the list of abbreviations for Table 2.
229) Please change "GI= gastrointestinal, NSAIDs= non-steroidal anti-inflammatory drugs, PM= poor metabolizer, IM= intermediate metabolizer, NM= normal metabolizer, UM= ultra metabolizer, UGIB= upper gastrointestinal bleeding, TXB2= Thromboxane B2" to "GI = gastrointestinal, NSAIDs = non-steroidal anti-inflammatory drugs, PM = poor metabolizer, IM = intermediate metabolizer, NM = normal metabolizer, UM = ultra metabolizer" (line 435).
230) Please replace "pain. [83]" with "pain [83]." (line 440).
231) Please change "fibromyalgia. [84]" to "fibromyalgia [84]." (line 441).
232) Please replace "Monoamine" with "monoamine" (line 443).
233) Please change "(MAOIs)" to "(MAOIs)," (line 444).
234) Please replace "antidepressants. (Table 3)" with "antidepressants (Table 3)." (line 444).
235) Please change "antidepressants (TCAs)" to "antidepressants" (line 445).
236) Please replace "hand" with "hand," (line 447).
237) Please change "treatment, such" to "treatment such" (line 449).
238) Please replace "TCAs" with "TCA" (line 452).
239) Please change "(EMA and FDA)" to "(EMA, www.ema.europa.eu; and FDA, www.fda.gov)" (line 456).
240) Please remove hyperlink from "EMA" and "FDA" in "(EMA and FDA)" (line 456).
241) Please remove underlined formatting from "EMA" and "FDA" in "(EMA and FDA)" (line 456).
242) Please replace "amitriptyline. [90]" with "amitriptyline [90]." (line 462).
243) Please change "randomized-controlled" to "randomized controlled" (line 462).
244) Please replace "Ryu et al., reported" with "Ryu et al. reported" (line 465).
245) "Ryu et al., reported" could read better as "This study" (line 465).
246) Please change "especially influenced by CYP2C19" to "influenced by CYP2C19 rather" (line 465).
247) Please replace "CYP2D6. [87]" with "CYP2D6 [87]." (line 467).
248) Please change "concentration. [91]" to "concentration [91]." (line 469).
249) It is not clear what the authors exactly mean by "slow metabolizers" and "fast metabolizers" in "A significantly lower pain intensity level was observed in CYP2D6 slow metabolizers compared to CYP2D6 fast metabolizers during the first week of treatment of postamputation pain in 30 patients initially receiving amitriptyline [92] (line 472) and in "Moreover, this study showed that CYP1A2 fast metabolizers probably experience more severe side effects compared to slow metabolizers. [120] (Table 3)" (line 579) as these categories are nowhere defined?
250) Please replace "normal metabolizers" with "NMs" (line 477).
251) Please change "IM" to "IMs" (line 477).
252) Please replace "studies above" with "abovementioned studies" (line 478).
253) Please change "researched. (Table 3)" to "researched (Table 3)." (line 480).
254) Please replace "NMs" with "NMs," (line 484).
255) "Steimer et al identified that patients carrying two functional CYP2D6 alleles combined with only one functional CYP2C19 allele showed the lowest risk of side effects compared to carriers of other combinations of alleles, especially compared to those with only one functional CYP2D6 allele as studied in 5o Caucasians with depressive disorder [94]" (line 485) is too long. Please split into two sentences.
256) Please change "5o" to "50" (line 488).
257) Please replace "[94]" with "[94]." (line 488).
258) Please change "investigated also" to "also investigated" (line 489).
259) Please replace "amitriptyline, but" with "amitriptyline but" (line 491).
260) Please change "association. [87](Table 3)" to "association (Table 3) [87]." (line 491).
261) Please replace "[88]" with "[88]." (line 496).
262) Please change "neuralgia" to "neuralgia," (line 500).
263) Please replace "metabolite. [94]" with "metabolite [94]." (line 502).
264) It is not clear what the authors mean by "tmax" in "This study detected two times higher tmax in the volunteers with two active OCT1 alleles (OCT1*1) comparing to those, who are carriers of one active allele (OCT1*1) and one inactive or reduced activity allele (OCT1*2,3*,4*) and subjects who carry two inactive or reduced activity alleles (OCT1*2, *3, *4, *5)" (line 506)?
265) Please change "morphine/nortriptyline" to "morphine and nortriptyline" (line 513).
266) Please replace "CYP2C19 AND" with "CYP2C19, and" (line 515).
267) Please remove italics formatting from "AND" (line 515).
268) Please change "nortriptyline/morphine. [96] (Table 3)" to "morphine and nortriptyline (Table 3) [96]." (line 517).
269) Please replace "et al." with "et al.," (line 518).
270) Please change "identified. [96] (Table 3)" to "identified (Table 3) [96]." (line 520).
271) Please replace "(if nortriptyline is warranted 50% reduction of the recommended starting dose) [88]" with "(50% reduction of the recommended starting dose if nortriptyline is warranted) [88]." (line 524).
272) Please change "off label" to "off-label" (line 527).
272) From "Imipramine is initially metabolized to desipramine by CYP2C19, with desipramine subsequently metabolized to a less active hydroxy-imipramine by CYP2D6. [98]" (line 528) is not clear whether desipramine is metabolized to 2-hydroxyimipramine or 10-hydroxyimipramine?
273) Please replace "CYP2C19, with" with "CYP2C19 with" (line 529).
274) Please change "hydroxy-imipramine by CYP2D6. [98]" to "hydroxyimipramine by CYP2D6 [98]." (line 530).
275) Please replace "NMs . [99,100]" with "NMs [99,100]." (line 532).
276) Please change "effects ," to "effects," (line 532).
277) Please replace "patients with CYP2D6 PM or CYP2C19 PM [98,101]" with "PM patients with CYP2D6 or CYP2C19 [98,101]." (line 534).
278) Please format "N" in "N-demethylation" using italics (line 535).
279) Please change "patients with genetic defects in the CYP2C19 gene (PM), hint" to "PM patients with genetic defects in the CYP2C19 gene, hinting" (line 536).
280) Please replace "imipramine. [102]" with "imipramine [102]." (line 537).
281) Please change "treatment, Siegenthaler" to "treatment. Siegenthaler" (line 540).
282) Please replace "detected. (Table 3)" with "detected (Table 3)." (line 544).
283) Please change "effects, including" to "effects including" (line 545).
284) Please replace "vivo. (Table 3)" with "vivo (Table 3)." (line 548).
285) Please change "inhibitors (SSRIs)" to "inhibitors" (line 549).
286) Please replace "Citalopram/Escitalopram" with "Citalopram and escitalopram" (line 557).
287) Please format "S" in "S-enantiomer" using italics (line 558).
288) Please change "CYP2D6. [113]" to "CYP2D6 [113]." (line 563).
289) Please replace "citalopram/escitalopram" with "citalopram and escitalopram" (lines 563, 565).
290) Please change "PMs" to "PMs," (line 564).
291) Please replace "events. [114–116]" with "events [114–116]." (line 565).
292) Please change "concentrations. [117]" to "concentrations [117]." (line 566).
293) Please replace "[118]" with "[118]." (line 569).
294) Please change "five genes, i.e. HTR2A, HTR2C, CYP2C19, and ABCB1, were" to "HTR2A, HTR2C, CYP2C19, and ABCB1" (line 569).
295) Please replace "Only for the HTR2C variant rs6318 a statistically significant difference was noticed," with "A statistically significant difference was noticed only for the HTR2C variant rs6318" (line 570).
296) Please change "limited, and" to "limited and" (line 572).
297) Please replace "studies. (Table 3)" with "studies (Table 3)." (line 573).
298) Please replace "disorders. [119]" with "disorders [119]." (line 575).
299) Please change "fatigue, and nausea/vomiting" to "fatigue, nausea, and vomiting" (line 576).
300) Please replace "al., detected" with "al. detected" (line 577).
301) Please change "rs4646425" to "rs4646425," (line 577).
302) Please replace "and nausea/vomiting, and fatigue" with "fatigue, nausea, and vomiting" (line 578).
303) Please change "metabolizers. [120] (Table 3)" to "metabolizers (Table 3) [120]." (line 580).
304) Please replace "citalopram. [121]" with "citalopram [121]." (line 583).
305) Please change "citalopram" to "citalopram," (line 584).
306) Please replace "citalopram. [122]" with "citalopram [122]." (line 585).
307) Please widen the "Variants" column in Table 3 so that "rs35796837," (Kuo et al. (2013) [112]) and "rs34058039," (Kuo et al. (2013) [112]) fit in one row.
308) Please change "efficacy" to "efficacy." (Chaudhry et al. (2017) [90]), "imipramine" to "imipramine." (Schliessbach et al. (2018) [101]), "HTR2A, HTR2C, ABCB1, CYP2C19, SLC6A4" to "HTR2A HTR2C ABCB1 CYP2C19 SLC6A4" (Brasch-An-dersen et al.(2011) [110]), "dry mouth, and nausea/vomiting" to "dry mouth, nausea, and vomiting" (Kuo et al. (2013) [112]) in Table 3.
309) Please add "FDA", "EMA", "TCA", "N/A", "SSRI" to the list of abbreviations for Table 3.
310) Please replace "PM= poor metabolizers, UM= ultra-metabolizer, IM= intermediate metabolizer, NM=normal metabolizer" with "PM = poor metabolizers, UM = ultra-metabolizer, IM = intermediate metabolizer, NM = normal metabolizer" (line 589).
311) Please change "for 9 drugs relevant medication-gene interactions" to "relevant medication-gene interactions for 9 drugs" (line 595).
312) Please replace "ABCC2" with "ABCC2," (line 597).
313) Please change "CYP1A2" to "CYP1A2," (line 597).
314) Please replace "NSAIDs" with "NSAIDs," (lines 599, 608).
315) Please change "on" to "of" (line 601).
316) Please replace "geno-" with "genotypes" (line 602).
317) Please change "the, to a lesser extent, studied" to "the less often studied" (line 603).
318) "Compared to the number of identified studies that investigated the impact of the mentioned CYP variants on the pharmacokinetics of NSAIDs the number of studies investigating the impact of these variants on pain relief are overall very limited, detecting only a very modest impact of CYP2C8 and CYP2C9 genetic variants on e.g. ibuprofen therapeutic effect" (line 606) is too long. Please split into two sentences.
319) Please replace "are" with "is" (line 609).
320) "on e.g. ibuprofen therapeutic effect" does not seem to fit well in "Compared to the number of identified studies that investigated the impact of the mentioned CYP variants on the pharmacokinetics of NSAIDs the number of studies investigating the impact of these variants on pain relief are overall very limited, detecting only a very modest impact of CYP2C8 and CYP2C9 genetic variants on e.g. ibuprofen therapeutic effect" (line 606). Please generalize the statement or aim its focus only on ibuprofen.
321) Please change "NSAIDs, and" to "NSAIDs and" (line 611).
322) "Studies detected here for several drugs, such as ibuprofen, diclofenac, or meloxicam a relation between slow metabolizer genotypes and an increased risk for the important side effects gastrointestinal bleeding or cardiovascular events" (line 612) does not seem to be grammatically correct with respect to "Studies detected here for several drugs, such as ibuprofen, diclofenac, or meloxicam a relation" and "for the important side effects gastrointestinal bleeding". Please revise.
323) Please replace "however" with "however," (line 617).
324) Please change "NSAIDs/aspirin" to "NSAID" (line 619).
325) "making it necessary to perform" could better read as "calling for" (line 622).
326) Please replace "nortriptyline/morphine" with "nortriptyline and morphine" (line 627).
327) "alterations in the CYP2C19 gene and in CYP2C19" does not seem to make sense in "Genetic alterations in the CYP2C19 gene and in CYP2C19 and CYP2D6 may be predictive for the risk of adverse effects for imipramine and amitriptyline, respectively, as explored in several studies including depressive patients" (line 627) as the CYP2C19 gene is mentioned twice. Please rephrase.
328) Please remove italics formatting from "SNP" (line 632).
329) "NSAIDs or antidepressants" could better read as "these drugs" (line 638).
330) Please change "I.I.E.,M.A.A. and" to "I.I.E., M.A.A., and" (lines 649, 650, 652, 653).
331) Please replace "methodology" with "methodology," (line 649).
332) Please change "J.M" to "J.M." (line 650).
333) Please replace "M.A.A ;" with "M.A.A.;" (lines 650, 651).
334) Please change "M.A.A" to "M.A.A." (lines 650 2x, 651, 652).
335) Please replace "J.M. ;" with "J.M.; (lines 652, 653).
336) Please change "VVT, VNC, HBS and JM" to "V.V.T., V.N.C., H.B.S., and J.M." (line 653).
337) Please replace "I.I.E.,M.A.A., HBS and" with "I.I.E., M.A.A., H.B.S., and" (line 654).
338) Please change "V.N.C., H.B.S.." to "V.N.C., and H.B.S." (line 655).
339) Please replace "manuscript.”" with "manuscript." (line 655).
340) Please translate "Vetenskapsrådet" into English language and provide relevant grant codes.
341) Please change "(HBS)" to "(H.B.S.)." (line 657).
342) Please ident the abbreviation list in the "Abbreviations" section so that individual terms are exactly aligned under each other.
343) Please replace "2," with "2" (line 665).
344) Please change "Normal" to "normal" (line 668).
345) Please replace "Nonsteroidal anti-inflammatory drugs," with "nonsteroidal anti-inflammatory drugs" (line 669).
346) Please change "PM. poor metabolizer," to "PM poor metabolizer" (line 670).
347) Please replace "polymorphisms," with "polymorphisms" (line 671).
348) Please change "inhibitors," to "inhibitors" (line 673).
349) Please replace "TNF-α." with "TNF-α" (line 674).
350) Please change "antidepressants," to "antidepressants" (line 675).
351) Please replace "Uridine 5'-diphospho-glucuronosyltransferase," with "uridine 5'-diphospho-glucuronosyltransferases" (line 676).
352) Please change "Upper" to "upper" (line 676).
353) Please replace "Vitamin" with "vitamin" (line 679).
Reviewer 4 Report
This review aims to elucidate the pharmacogenetic evidence and highlight genes that play a role in the pharmacodynamics and safety of anti-inflammatory and antidepressant drugs used in pain management. The choice of subjects makes sense. The comments on how to select the literature are detailed. The results can be used as a reference. I think this review can be published.
Some points:
The topic of the paper seems to be more relevant to the journal "Pharmacogenomics".
The title need to highlight the related topics about non-steroidal anti-inflammatory drugs (NSAIDs) and antidepressants.
This can be further simplified in the section describing "2. Pain management".
Please check other literature on the subject for relevance, eg Drug Metabolism and Personalized Therapy 35(3); Pharmacogenomics 22(9), pp. 573-586.
Line 636 well-powered
Line 644 heart-related
Author Response
This review aims to elucidate the pharmacogenetic evidence and highlight genes that play a role in the pharmacodynamics and safety of anti-inflammatory and antidepressant drugs used in pain management. The choice of subjects makes sense. The comments on how to select the literature are detailed. The results can be used as a reference. I think this review can be published.
Some points:
- The topic of the paper seems to be more relevant to the journal "Pharmacogenomics".
Answer of the authors:
Thank you for the remark. The note regarding alternative journal options is acknowledged.
- The title need to highlight the related topics about non-steroidal anti-inflammatory drugs (NSAIDs) and antidepressants. This can be further simplified in the section describing "2. Pain management".
Answer of the authors:
Thank you for the remark. The title has been changed to:
Pharmacogenetics and pain treatment with a focus on non-steroidal anti-inflammatory drugs (NSAIDs) and antidepressants: a systematic review
- Please check other literature on the subject for relevance, eg Drug Metabolism and Personalized Therapy 35(3); Pharmacogenomics 22(9), pp. 573-586.
Answer of the authors:
Thank you for the suggestion, we have added information from the mentioned paper on page 2 lines 92 in the introduction.
- Line 636 well-powered
Answer of the authors:
We have corrected the spelling accordingly.
- Line 644 heart-related
Answer of the authors:
We have corrected the spelling accordingly.
Round 2
Reviewer 3 Report
Zobdeh et al. have prepared a comprehensive systematic review on the pharmacogenetic aspects of nonsteroidal anti-inflammatory drugs and antidepressants against pain in human. Their manuscript provides a unique insight into the complexity of gene-drug interactions while highlighting that these relationships are seldomly straightforward. The screened articles are neatly summarized in accompanying tables, which nicely encapsulate the derived conclusions as well as help to easily signpost the reader to relevant references. In conclusion, the work by Zobdeh et al. will represent a useful resource for future generations of clinicians as well as researchers in pain therapy.
1) Please change "antidepressants :" to "antidepressants:" (line 4).
2) Please format "CYP2C9" using italics (lines 30, 31 3x, 32 2x, 488, 492, 524, 525, 620, 740, 741, 1289, 1333).
3) Please format "CYP2C8" using italics (line 31 2x).
4) Please format "UGT2B7" using italics (line 31).
5) Please format "ABCC2" using italics (line 31).
6) Please format "SLCO1B1" using italics (line 32).
7) Please format "CHST2" using italics (line 32).
8) Please format "CYP2D6" using italics (lines 32, 939, 940, 941, 1019, 1020, 1021, 1044 2x, 1045 2x, 1055, 1317, 1330).
9) Please format "CYP2C19" using italics (lines 32, 33 3x, 939, 1019, 1038, 1044, 1046, 1055, 1057, 1067, 1068, 1166, 1171, 1181 2x, 1185, 1187, 1189, 1190 2x, 1330).
10) Please format "ABCB1" using italics (lines 33, 1171).
11) Please format "HTR2C" using italics (lines 33, 1170, 1172).
12) Please format "CYP1A2" using italics (line 34).
13) Please replace "anti-inflammatory acting" with "anti-inflammatory" (line 40).
14) Please replace "(IASP) (www.iasp-pain.org)and" with "(IASP, www.iasp-pain.org) and" (line 56).
15) Please change "(PM), as a result" to "(PM). As a result," (line 88).
16) Please replace "effects" with "effects," (line 88).
17) Please change "review has" to "reviews have" (line 89).
18) Please replace "neuropathic" with "such as neuropathic" (line 148).
19) Please replace "fibromyalgia" with "fibromyalgia," (line 148).
20) Please change "or" to "or with" (line 150).
21) Please replace "(e.g. pamidronate)" with "(pamidronate)" (line 154).
22) Please change "morphine" to "or morphine" (line 161).
23) "benefits and risks should be considered in case that increasing dosage is needed" in "Opioids should be added to treatment gradually with the lowest effective dose and may be used in combination with first-stage medications (NSAIDs and paracetamol can potentiate the effects of opioids) benefits and risks should be considered in case that increasing dosage is needed" (line 161) sounds rather vague. What benefits and risks should be considered?
24) Please change "(NSAIDs and paracetamol can potentiate the effects of opioids) benefits" to "(NSAIDs and paracetamol can potentiate the effects of opioids). Benefits" (line 163).
25) Please replace "fentanyl" with "fentanyl," (line 165).
26) Please change "1 WHO analgesic Ladder" to "1. WHO analgesic ladder" (line 255).
27) Please replace "links" with "link" (line 266).
28) Please change "“ UGT”" to "“UGT”" (line 274).
29) Please remove italic formatting from "UGT" (line 274).
30) Please remove italic formatting from "CYP2C9" (lines 274, 386, 496, 626, 727, 910).
31) Please remove italic formatting from "CYP2C8" (lines 275, 386).
32) Please remove italic formatting from "CYP2C19" (lines 275, 903, 909, 1051, 1052, 1158).
33) Please remove italic formatting from "CYP2B6" (line 275).
34) Please remove italic formatting from "CYP2D6" (lines 275, 903, 910, 1028, 1051, 1052, 1158).
35) Please remove italic formatting from "CYP1A2" (lines 275, 910, 1051).
36) Please remove italic formatting from "CYP3A4" (lines 275, 386, 497, 727, 910, 1051).
37) Please remove italic formatting from "CYP3A5" (line 275).
38) Please remove italic formatting from "COMT" (line 276).
39) Please remove italic formatting from "ABCB1" (line 276).
40) Please remove italic formatting from "SLC6A4" in "“SLC6A4, SERT”" (line 276).
41) Please remove italic formatting from "SLC6A2" in "“NET, SLC6A2”" (line 276).
42) Please replace "(therapy efficacy or safety), highlighting" with "(therapy efficacy or safety) highlighting" (line 318).
43) Please change "are" to "were" (line 319).
44) Please replace "Tetracyclic antidepressant" with "Tetracyclic antidepressants" in Table 1.
45) Please change "studies included in review" to "Studies included in review" in Figure 2.
46) Please replace "pharmacokinetics" with "pharmacokinetic." (line 372).
47) Please change "Cytochrome" to "cytochrome" (line 385).
48) Please remove italic formatting from "UGT2B7" (line 387).
49) Please remove italic formatting from "UGT1A6" (lines 387, 747).
50) Please remove italic formatting from "UGT1A4" (line 387).
51) Please replace "UGT1A4" with "UGT1A4," (line 387).
52) Please remove italic formatting from "UGT1A9" (lines 387, 747).
53) Please remove italic formatting from "CYP2C9" (line 396).
54) Please remove italic formatting from "CYP2C8" (line 396).
55) Please change "hours" to "hr" (lines 408).
56) Please replace "in CYP2C9 PMs compared with intermediate (IM) or normal (NM) metabolizers was observed" with "was observed in CYP2C9 PMs compared with intermediate (IM) or normal (NM) metabolizers" (line 409).
57) From "In the frame of the study, the authors reported multiple adverse effects including hepatic profile alteration, acute rhabdomyolysis, aural skin rash, headache, and abdominal pain in 7 out of 122 volunteers" (line 419) is not clear whether "7 out of 122 volunteers" had "abdominal pain" or "multiple adverse effects including hepatic profile alteration, acute rhabdomyolysis, aural skin rash, headache, and abdominal pain"?
58) Please change "system" to "system," (line 489).
59) Please replace "are PM, individuals with a score between 1 to 1.5 are classified as" with "are classified as PMs, individuals with a score between 1 to 1.5 are" (line 489).
60) Please change "IM and those with a score of 2 are NM" to "IMs and those with a score of 2 are NMs" (line 490).
61) "Although, Ustare et al. reported that response to celecoxib was better for postoperative pain in 2 patients among 99 IM patients with CYP2C9*1/CYP2C9*3 genotype compared to UM and PM patients (Table 2)" (line 515) does not seem to be grammatically correct with respect to "Although". Please revise.
62) Please format "CYP2C9*1/CYP2C9*3" using italics (line 516).
63) Please change "report" to "report," (line 520).
64) Please replace "to stepwise increase the dose" with "increase the dose in a stepwise fashion" (line 523).
65) Please replace "PMs" with "PMs," (line 525).
66) Please replace "al" with "al." (lines 613, 650, 1170).
67) "Subjects reported sleepiness (carrying CYP2C8*3 mutant and CYP2C9*1/*3 genotype) and stomachaches (carrying CYP2C8*3 mutant and the CYP2C9*1/*3 genotype)" (line 616) contrasts with "Calvo et al [54], who studied the relation between CYP2C8 and CYP2C9 genotypes and the clinical efficacy of oral piroxicam as mentioned above, detected two subjects among 102 individuals with adverse reactions with one carrying the CYP2C8*3 mutant and the CYP2C9*1/*3 genotype" (line 613) as only one subject carrying the CYP2C8*3 mutant and the CYP2C9*1/*3 genotype is mentioned in the latter sentence.
68) Please change "Subjects reported sleepiness (carrying CYP2C8*3 mutant and CYP2C9*1/*3 genotype) and stomachaches (carrying CYP2C8*3 mutant and the CYP2C9*1/*3 genotype)" to "Subjects carrying the CYP2C8*3 mutant and the CYP2C9*1/*3 genotype reported sleepiness and stomachaches" (line 616).
69) It is not clear what the authors mean by "the drug" in "To date, no in vivo study was found that evaluated genetic variations in enzymes involved in ketoprofen metabolism and their effect on pain reduction with the drug" (line 634)?
70) Please provide reference for "Mejía-Abril et al. reported no association between genetic polymorphisms of CYP1A2, CYP2A6, CYP2B6, CYP2C8, CYP2C9, CYP2C19, CYP2D6, CYP3A4, CYP3A5, UGT1A1, ABCB1, ABCC2, SLCO1B1, and SLC22A1 and adverse effects of dexketoprofen" (line 639).
71) Please replace "is acting" with "acts" (line 645).
72) Please change "date" to "date," (line 654).
73) Please provide reference for "Daly et al. studied the association between the genetic predisposition and diclofenac-induced hepatotoxicity in 24 patients who had suffered from diclofenac hepatotoxicity" (line 657).
74) It is not clear what the authors mean by "lowest recommended" and "starting by 50%" in "The CPIC recommends for meloxicam treatment of CYP2C9 IMs with activity score 1 to reduce the lowest recommended starting by 50% and to carefully dose titrate until steady state is reached" (line 739)?
75) Please remove italic formatting from "UGT1A1" (line 747).
76) Please remove italic formatting from "UGT1A7" (line 747).
77) Please replace "Aspirin" with "aspirin" (line 756).
78) Please provide reference for "Shiotani et al. detected that carriers of the SLCO1B1*1b haplotype and the CHST2 2082 T allele were at significantly higher risk for peptic ulcer and ulcer bleeding compared to controls when using aspirin" (line 756).
79) Please change "(n = 45) and control group (n = 482)" to "(n = 45), and a control group (n = 482)," (line 759).
80) Please replace "take" with "took" (line 765).
81) Please replace "TNF-α -863C> A" with "TNF-α -863C> A," (line 766).
82) Please provide reference for "Groza et al. tested possible correlation between VKORC1 -1639 G>A polymorphism and UGIB among 163 patients diagnosed with UGIB" (line 770).
83) Please insert a gap between line 856 and Table 2.
84) It is not clear what the authors mean by "on 24th and 48th hours" in "Lower pain scores in CYP2C9 IMs on 24th and 48th hours compared to NMs" in Table 2 ((Ustare et al. (2020))?
85) Similarly, it is not clear what the authors mean by "i" in "(UGIB)i" in Table 2 (Wang et al. (2019))?
86) Please change "(n= 124 )" to "(n = 124 )" (Martinez et al. (2004)), "Ums" to "UMs" (Saiz-Rodriguez et al. (2021)), "hours" to "hr" (Saiz-Rodriguez et al. (2021), Ustare et al. (2020)), "Malay-Polynesian , Filipinos" to "Malay-Polynesian, Filipinos" (Ustare et al. (2020)), "n=28/48" to "n = 28/48" (Daly et al. (2007)), "112" to "n = 112" (Daly et al. (2007)), "ABCC2 C-24/C-24 ,C-24/T-24 ,T-24/T-24" to "ABCC2 C-24/C-24, C-24/T-24, T-24/T-24" (Daly et al. (2007)), "n=24" to "n = 24" (Aithal et al. (2000)), "n=100" to "n = 100" (Aithal et al. (2000)), "n=111" to "n = 111" (Shiotani et al. (2014)), "n=45" to "n = 45" (Shiotani et al. (2014)), "n=482" to "n = 482" (Shiotani et al. (2014)), "alleles" to "allele" (Shiotani et al. (2014)), "n=57" to "n = 57" (Wang et al. (2019)), "n=97" to "n = 97" (Wang et al. (2019)), "- 863C" to "-863C" (Wang et al. (2019)), "n=154" to "n = 154" (Groza et al. (2017)), "n=178" to "n = 178" (Groza et al. (2017)), "n=88" to "n = 88" (E. Piazuelo et al. (2008)), "n=108" to "n = 108" (E. Piazuelo et al. (2008)), "hematemesis , melena" to "hematemesis, melena" (Figueiras et al. (2016)), "n=577" to "n = 577" (Figueiras et al. (2016)), "n=1343" to "n = 1343" (Figueiras et al. (2016)), "DexKetoprofen" to "Dexketoprofen" (Mejía-Abril et al. (2021)) in Table 2.
87) Please format "CYP2C8" (Saiz-Rodriguez et al. (2021), Weckwerth et al. (2020)), "CYP2C9" (Saiz-Rodriguez et al. (2021) 2x, Weckwerth et al. (2020), Jaja et al. (2015), Hamilton et al. (2020), Ustare et al. (2020)), "CYP2C19" (Saiz-Rodriguez et al. (2021)), "CYP2D6" (Saiz-Rodriguez et al. (2021)) using italics in Table 2.
88) Please remove bold formatting from "(" in "(possible differences in the incidence of cardiovascular complications and bleeding)" in Table 2 (Lee et al. (2014)).
89) Please define abbreviation for "ia" in Table 2 (Figueiras et al. (2016)).
90) Please replace "B2" with "B2." (line 884).
91) Please change "depression, and" to "depression and" (line 899).
92) Please replace "(EMA, www.ema.europa.eu; and FDA, www.fda.gov)" with "(European Medicines Agency (EMA), www.ema.europa.eu; and Food and Drug Administration (FDA), www.fda.gov)" (line 908) and "European Medicines Agency (EMA) or Food and Drug Administration (FDA)" with "EMA or FDA" (line 1025).
93) Please provide reference for "Ryu et al. performed a randomized controlled trial including 24 healthy adults with the aim to study the pharmacokinetics of amitriptyline in relation to the genotypes of CYP2C19 (CYP2C19*2/*2, *2/*3, or *3/*3) and CYP2D6 (CYP2D6*10/*10)" (line 913).
94) Please remove italic formatting from "and" (line 915).
95) Please change "further researched" to "researched further" (line 929).
96) Please provide reference for "Chaudhry et al. reported a trend toward more severe adverse effects in patients with diabetic peripheral neuropathy with lower CYP2D6 activity scores" (lie 930).
97) Please provide reference for "Steimer et al. identified that patients carrying two functional CYP2D6 alleles combined with only one functional CYP2C19 allele showed lower risk of side effects compared to carriers of other combinations of alleles" (line 933).
98) Please replace "Ums" with "UMs" (lines 940, 1175).
99) Please change "UM" to "UMs" (lines 1020, 1186).
100) Please replace "IM and PM" with "IMs and PMs" (line 1021).
101) Please change "dose" to "dose, respectively" (line 1021).
102) Please replace "pain. [97]" with "pain [97]." (line 1026).
103) Please replace "pharmacokinetic" with "pharmacokinetics" (line 1029).
104) Please change "higher time of the" to "higher" (line 1031).
105) Please provide reference for "Benavides et al. examined the association between genetic markers and the analgesic effect of the combination therapy with morphine and nortriptyline or the respective monotherapies in patients with neuropathic pain" (line 1036).
106) Please replace "to" with "to the" (line 1044).
107) Please replace "(25% reduction of the recommended starting dose )" with "(25% reduction of the recommended starting dose)" (line 1045).
108) Please change "(50% reduction of the recommended starting dose if nortriptyline is warranted)" to "(50% reduction of the recommended starting dose if nortriptyline is warranted)," (line 1045).
109) Please replace "is used" with "also" (line 1050).
110) Please change "CYP2D6. [100]" to "CYP2D6 [100]." (line 1052).
111) Please replace "SNP" with "SNPs" (line 1057).
112) Please change "[104] observed that imipramine showed no significant effect on low back pain reduction in 50 patients in general" to "observed that imipramine showed no significant effect on low back pain reduction in 50 patients in general [104]" (line 1059).
113) Please change "indication" to "indication of" (line 1164).
114) Please replace "adjusting" with "decreasing" (line 1067).
115) Please replace "PM" with "PMs" (lines 1067, 1179, 1187).
116) Please change "UM and IM" to "UMs and IMs" (line 1068).
117) Please replace "is" with "were" or "have been" (line 1157).
118) Please replace "and to a lesser extent by" with "and, to a lesser extent, by" (line 1164).
119) Please format "CYP2C1" in "CYP2C19" using italics (line 1167).
120) Please change "genes" to "gene" (line 1169).
121) Please format "HTR2A" using italics (line 1170).
122) Please replace "nausea" with "nausea," (lines 1177, 1178).
123) Please provide reference for "Kuo et al. detected that CYP1A2 SNPs rs2069521, rs4646425, and rs4646427 have a notable relation with dry mouth, fatigue, nausea and vomiting" (line 1177).
124) Please change "RM" to "RMs" (line 1179).
125) Please replace "9777" with "9,777" (line 1183).
126) Please define abbreviation for "ECG" (line 1183) and "ADs" (line 1285) and these to the Abbreviations section.
127) Please change "leads to the recommendation to prescribe" to something like "necessitates the prescription of" (line 1188).
128) Please replace "IM" with "IMs" (line 1189, 1190).
129) Please change "contrast to this" to "contrast" (line 1190).
130) Please change "metabolizer (EM)" with "metabolizers (EMs)" (line 1190).
131) Please replace "Brasch-Andersen et al.(2011)" with "Brasch-Andersen et al. (2011)" (Brasch-Andersen et al.(2011)), "CYP2C19 ," with "CYP2C19," (Brasch-Andersen et al.(2011)), "nausea" with "nausea," (Kuo et al. (2013)) in Table 3.
132) Please format "CYP2D6" using italics in Table 3 (Chaudhry et al. (2017)).
133) Please change "metabolizers" to "metabolizer" (line 1274).
134) Please replace "inhibitor" with "inhibitor." (line 1276).
135) Please change "poor metabolizer" to "PM" (lines 1288, 1333).
136) Please change "diclofenac," to "diclofenac, and" (line 1291).
137) Please replace "NSAIDs" with "NSAID" (line 1306).
138) Please change "drugs, such" to "drugs such" (line 1308).
139) Please change "limited" to "limited," (line 1316).
140) Please replace "[94], and escitalopram adverse effects [120], include" with "[94] and escitalopram adverse effects [120] include" (line 1336).
141) Please change "review, mentioned" to "review mentioned" (line 1344).
142) Please replace "articles, were" with "articles were" (line 1345).
143) Please change "emphasises" to "emphasizes" (line 1346).
144) Please change "“Conceptualization," to "Conceptualization," (line 1375).
145) Please change "M.A.A" to "M.A.A." (line 1376).
146) Please replace "J.M" with "J.M." (line 1379).
147) Please change "manuscript.”" to "manuscript." (line 1380).
148) Please replace "2019-01066" with "2019-01066." (line 1384).
